# Bayesian Modeling and Uncertainty Quantification for Learning to Optimize: What, Why, and How

**Yuning You**[1], **Yue Cao**[1], **Tianlong Chen**[3], **Zhangyang Wang**[3], **Yang Shen**[1,2]
[1]Department of Electrical and Computer Engineering, Texas A&M University
[2]Department of Computer Science and Engineering, Texas A&M University
[3]Department of Electrical and Computer Engineering, University of Texas at Austin
{yuning.you,cyppsp,yshen}@tamu.edu, {tianlong.chen,atlaswang}@utexas.edu

## Abstract

Optimizing an objective function with uncertainty awareness is well-known to improve the accuracy and confidence of optimization solutions. Meanwhile, another relevant but very different question remains yet open: how to model and quantify the uncertainty of an optimization algorithm (a.k.a., optimizer) itself? To close such a gap, the prerequisite is to consider the optimizers as sampled from a distribution, rather than a few prefabricated and fixed update rules. We first take the novel angle to consider the algorithmic space of optimizers, and provide definitions for the optimizer prior and likelihood, that intrinsically determine the posterior and therefore uncertainty. We then leverage the recent advance of learning to optimize (L2O) for the space parameterization, with the end-to-end training pipeline built via variational inference, referred to as uncertainty-aware L2O (**UA-L2O**). Our study represents the first effort to recognize and quantify the uncertainty of the optimization algorithm. The extensive numerical results show that, UA-L2O achieves superior uncertainty calibration with accurate confidence estimation and tight confidence intervals, suggesting the improved posterior estimation thanks to considering optimizer uncertainty. Intriguingly, UA-L2O even improves optimization performances for two out of three test functions, the loss function in data privacy attack, and four of five cases of the energy function in protein docking. Our codes are released at https://github.com/Shen-Lab/Bayesian-L2O.

## 1 Introduction

Computational models of many real-world applications involve optimizing non-convex objective functions. As the non-convex optimization problem is NP-hard, no optimization algorithm (or optimizer) could guarantee the global optima in general. Instead their solutions' usefulness (sometimes based on their proximity to the optima), when the optima are unknown, can be very uncertain. Being able to quantify solution uncertainty directly provides calibration with ensured awareness of the solution quality and usefulness (and another potential benefit is in optimization performance by enhancing the search efficiency). For instance, reliable and trustworthy machine learning models demand uncertainty awareness and quantification (UQ) during training (optimizing) such models, whereas in reality deep neural networks without proper modelling of uncertainty suffer from over-confidence and miscalibration (Guo et al., 2017). In another example of 3D prediction of protein-protein interactions, even though there exists the model uncertainty of the objective function and the data uncertainty of the protein structure data (Cao & Shen, 2020), state-of-the-art methods only predict several ranked solutions (Porter et al., 2019) without any associated uncertainty, which is hard for biologists to interpret.

Despite progress in optimization with uncertainty-awareness, fundamental open questions remain: existing methods consider uncertainty either within the data or the model (including objective functions) (Kendall & Gal, 2017; Ortega et al., 2012; Lyu et al., 2021; Cao & Shen, 2020), whereas inconspicuous attention was paid to the uncertainty arising from the **optimizer** that is directly re-

sponsible for deriving the end solutions. The optimizer is usually prefabricated and fixed. For instance, there are several popular update rules in Bayesian optimization, such as expected improvement (Vazquez & Bect, 2010) and upper confidence bound (Srinivas et al., 2009), that are chosen and unaltered for the entire process. For Bayesian neural networks training, the update rule of the iterative optimizer is usually chosen off-the-shelf, such as Adam, SGD, and RMSDrop. In principle, it is important to recognize the existence of an optimization algorithm **space** where a specific optimizer lies as well as the importance of the optimizer uncertainty, intrinsically defined in the space to the optimization and UQ solutions. In practice, such uncertainty is unwittingly omitted in the current context where an optimizer is treated as a *de facto* specified sample in the space.

A naïve characterization for a hand-built optimizer is to rely on a few of its hyper-parameters (Wenzel et al., 2020; Lorraine & Duvenaud, 2018), whereas such a parameterization space could be too biased and restricted to span a broader and more complicated optimizer manifold, potentially resulting in inaccurate UQ results (see Section 4, confidence estimation for test functions). Fortunately, the recent surge of learning to optimize (L2O) (Andrychowicz et al., 2016a; Li & Malik, 2016; Cao et al., 2019a; You et al., 2020; Li et al., 2020; Chen et al., 2020b;a; Shen et al., 2021; Chen et al., 2021) which is another prominent optimization paradigm, parameterized by a neural network using high-dimensional weights and following the data-driven ethos to learn the update rule adaptively, therefore introducing the less inductive bias and accompanied with the universal approximation ability, indicates a plausible solution to depict the sophisticated and more complete optimizer space.

**Contributions.** The aforementioned introduction reveals and discusses the fundamental question of why modelling the optimizer uncertainty, and our contingent questions would be, what defines the optimizer uncertainty and how to enable UQ during optimization. We give the following answers. (i) We define the prior and likelihood of the optimizer (Neal, 2012), which routinely determine the optimizer posterior with the general product rule and Bayes' theorem. The optimizer uncertainty is therefore well-defined. (ii) To unseal UQ within optimization, we treat an optimizer as a random sample from an algorithmic space of iterative optimizers. We leverage the surging L2O technique (Andrychowicz et al., 2016a; Li & Malik, 2016) with high-dimensional weights for the optimizer space parameterization. (iii) We further construct the end-to-end training pipeline via variational inference (Kingma & Welling, 2013; Higgins et al., 2016) to avoid the expensive computational cost of Markov chain Monte Carlo (MCMC) and degenerated posterior. The newly trained L2O is referred to as uncertainty-aware L2O (**UA-L2O**). In summary, the core intellectual value of this work is, for the first time, we recognize and quantify a novel form of uncertainty that lies in the optimization algorithm space parameterized by L2O, apart from the classical data- or model-based uncertainties (also known as epistemic and aleatoric uncertainties (Fox & Ülkümen, 2011)).

Extensive experiments show that UA-L2O had superior capability in uncertainty calibration thanks to the accurate estimation of solution posteriors. It confidence levels well matched the chance for the global optimum falling in the corresponding intervals. Intriguingly, although not directly targeted, UA-L2O also outperformed competing methods in optimization performance, as seen in optimizing two out of three test functions in high dimensions, the loss function in data privacy attack, energy functions in four out of five protein-docking cases.

## 2 PRELIMINARIES

### 2.1 LEARNING TO OPTIMIZE

Let us consider an optimization problem: $\min_{\mathbf{x}} f(\mathbf{x})$ where $\mathbf{x} \in \mathbb{R}^d$. A classic optimizer often iteratively updates $\mathbf{x}$ based on a handcrafted rule. For example, the first-order gradient descent algorithm takes an update at iteration $t$ based on the local gradient at the instantaneous point $\mathbf{x}_t$: $\mathbf{x}_{t+1} = \mathbf{x}_t - \alpha \nabla f(\mathbf{x}_t)$, where $\alpha$ is the step size.

Learning to Optimize (L2O) has lots of freedom to use the available information. We define $\mathbf{z}_t$ as optimization trajectories' historical information up to time $t$, e.g., the existing iterates $\mathbf{x}_0, \ldots, \mathbf{x}_t$, and/or their gradients $\nabla f(\mathbf{x}_0), \ldots, \nabla f(\mathbf{x}_t)$. L2O models an update rule by a predictor function $g$ of $\mathbf{z}_t$: $\mathbf{x}_{t+1} = \mathbf{x}_t - g(\mathbf{z}_t; \phi)$, where the mapping of $g$ is parameterized by $\phi$. Finding an optimal update rule can be formulated mathematically as searching for a good $\phi$ over the parameter space of $g$. Practically, $g$ is often a neural network. Since neural networks are universal approximators, L2O has the potential to discover completely new update rules without relying on existing rules. In order to find a desired $\phi$ associated with a fast optimizer, (Andrychowicz et al., 2016b) proposed to

minimize the weighted sum of the objective function $f(\mathbf{x}_t)$ over a time span $T$:

$$\min_{\boldsymbol{\phi}} \sum_{t=1}^{T} w_t f(\mathbf{x}_t), \quad \text{with} \quad \mathbf{x}_{t+1} = \mathbf{x}_t - g(\mathbf{z}_t; \boldsymbol{\phi}), \ t = 0, \ldots, T-1, \tag{1}$$

where $w_1, \ldots, w_T$ are the weights whose choices vary from case to case and depend on empirical settings. Note that $\boldsymbol{\phi}$ determines the objective value through determining the iterates $\mathbf{x}_t$. L2O solves the problem (1) for a desirable $\boldsymbol{\phi}$ and correspondingly the update rule $g(\mathbf{z}_t; \boldsymbol{\phi})$.

A typical L2O workflow is divided into two stages (Chen et al., 2021): a meta-training stage that learns the optimizer with a set of similar optimizees from the task distribution; and a meta-testing stage that applies the learned optimizer to new unseen optimizees. The meta-training process often occurs offline and is time consuming. However, the online application of the method, meta testing, is (aimed to be) time saving.

## 2.2 THE UNCERTAINTY OF OPTIMIZATION

The choice of optimizers is recognized to remarkably impact the solution quality, especially for non-convex and rugged objectives such as various loss functions for training deep networks. While non-convex optimization is NP-hard and the global optimum is never guaranteed, domain users usually can only blindly take the solutions given by a particular optimizer, or choose from several well-known algorithms based on human experiences or on an *ad hoc* basis. Such trial-and-error selections are biased and restricted to span a more complete optimizer manifold, and computationally expensive, even intractable when new algorithms are to be discovered by L2O, whose "fitness" or confidence on certain problem instances (especially instances with shifts from the target task distributions) is never known. Given a specific problem instance, if we can provide a confidence score or quantify the uncertainty of each candidate optimizer, it will certainly improve the selection and calibration of optimization algorithms, in particular the learned optimizers since they are essentially all "new", and hence enhancing the reliability of L2O.

As for non-convex objectives, different choices of optimizers, even each algorithm's hyper-parameter or initialization variations, can lead to vastly different solutions, implying a new type of uncertainty arising from choosing optimizers. Stochastic optimization methods like random search (Zhigljavsky, 2012), simulated annealing (Kirkpatrick et al., 1983) and genetic algorithms (Goldenberg, 1989) inject the randomness into algorithms to reduce solution uncertainty. However, they cannot reliably quantify its effect on the final solutions. Several works explored uncertainty quantification during optimization, by Bayesian optimization (Hennig & Schuler, 2012; Wang & Jegelka, 2017) or Monte Carlo sampling (Ortega et al., 2012). Several works in Bayesian optimization explored uncertainty quantification during optimization, by modeling the distributions over the functional space (Hennig & Schuler, 2012; Wang & Jegelka, 2017) or directly over the optimum (Ortega et al., 2012; Cao & Shen, 2020). However, those methods are often computationally too expensive, not to mention applied within the already costly L2O process. More importantly, they neither explicitly consider the uncertainty within the optimizer itself, nor concern the automatic learning and discovery of the most suitable optimizer.

## 3 TECHNICAL APPROACH

To quantify and address such "fitness" especially for the L2O parameterization in a principled way, we will introduce this new **optimizer uncertainty** arising from the choice of optimization algorithms, which is different from the classical data- or model-based uncertainties (also known as epistemic and aleatoric uncertainties (Fox & Ülkümen, 2011)). We will explore: how to quantify optimizer uncertainty? how to design a learned optimizer so that its uncertainty can be more accurately quantified? what is the best calibrated optimizer one can choose or learn for a given problem?

The core innovation is to treat an optimizer as a random sample from a continuous algorithmic space, rather than one of a few hand-crafted update rules, so as to model the intrinsic uncertainty within the optimizer. This novel view of algorithmic space is particularly enabled by L2O. Note that an L2O update rule is typically parameterized by a neural network $g$, with its inputs $\mathbf{z}_t$ and learnable parameters $\boldsymbol{\phi}$. Its learning capacity allows us to "sample" optimizers by taking $g(\cdot; \boldsymbol{\phi})$ with different weights $\boldsymbol{\phi}$. That is sufficiently versatile and flexible in practice.

### 3.1 Concepts for Optimizer Uncertainty

Without loss of generality, we define an iterative algorithmic space $\mathcal{G}$, where each point $g \in \mathcal{G}$ maps to an update $g(\mathbf{z}_t)$ dependent on the current/past zero-th order and/or first-order information $\mathbf{z}_t$. We treat $g$ as a random vector from $\mathcal{G}$, that leads to defining the optimizer uncertainty as follows:

**Definition 1 (Optimizer Uncertainty)** *Let $\mathcal{G}$ be the algorithmic space, where each point $g \in \mathcal{G}$ is an optimizer (omitting $\phi$ parameterization). We assume that*

1. *$g$ has a prior distribution $p(g)$;*

2. *Its likelihood can be interpreted as $p(\mathbf{z}_t|\mathbf{z}_{t_0}, g) = \prod_{i=t_0+1}^{t} p(\mathbf{x}_i|\mathbf{z}_{i-1}, g)$, $\forall t_0 < t$.*

The likelihood factorization is based on the causality of iterative optimizers, i.e., $(\mathbf{x}_i, \mathbf{z}_i)$ depends solely on $\{(\mathbf{x}_{i_0}, \mathbf{z}_{i_0})\}_{i_0 < i}$ given $g$. Thereby, we define the optimizer uncertainty at step $t$ as the posterior of $g$, that is a conditional distribution on the $t$ steps of the trajectory, using Bayes' theorem:

$$p(g|\mathbf{z}_t) \propto p(g) \prod_{i=1}^{t} p(\mathbf{x}_i|\mathbf{z}_{i-1}, g). \tag{2}$$

Intuitively, the prior $p(g)$ represents the belief about well-performing optimizers, the likelihood $p(\mathbf{z}_t|\mathbf{z}_{t_0}, g)$ represents the probability of observing an optimization trajectory (data) under a given optimizer $g$, and the posterior $p(g|\mathbf{z}_t)$ represents the probability for $g$ being the optimizer generating the observed data.

Prior work on hyper-parameter optimization (HPO) for classical algorithms (Feurer & Hutter, 2019) could be viewed as a special case of L2O (e.g., only a few hyper-parameters are "learnable", which results in a much biased and restricted prior $p(g)$). The seminal work Bergstra et al. (2011) optimized hyper-parameters using posterior-based fitness modeling. Recently, Wenzel et al. (2020) proposed a non-Bayesian approach of hyper-parameter ensembling that can estimate the predictive uncertainty of an optimization algorithm by varying hyper-parameters. Our work generalizes from tuning a few hyper-parameters to learning the entire optimizer space, and extends a Bayesian treatment. Since L2O involves much higher-dimensional parameters compared to HPO, its uncertainty quantification calls for computationally tractable and scalable approaches, which is detailed next.

### 3.2 L2O with Bayesian Uncertainty Quantification

Now that we have the optimizer space $\mathcal{G}$ parameterized in $\phi$, an optimizer $g$ is modeled as a random vector in the space. That allows for reducing the modeling of optimizer uncertainty $p(g|\mathbf{z}_t)$ to modeling the posterior $p(\phi|\mathbf{z}_t)$. For simplicity, we use a Gaussian prior of $\phi$: $p(\phi) \propto \exp(-\frac{1}{2\lambda}\|\phi\|_2^2)$ where $\lambda$ is a constant. We also follow the widely-adopted idea in (Gal & Ghahramani, 2016; Bishop & Tipping, 2003; Ortega et al., 2012; Cao & Shen, 2020), to view the loss function value as proportional to the negative logarithmic likelihood, i.e., $p(\mathbf{x}_t|\mathbf{z}_{t-1}, g) = p(\mathbf{x}_t|\mathbf{z}_{t-1}, \phi) \propto \exp(-f(\mathbf{x}_t^{\phi}))$. We can then calculate the posterior of the optimizer based upon equation (2):

$$p(\phi|\mathbf{z}_T) \propto \exp(-\frac{1}{2\lambda}\|\phi\|_2^2) \prod_{t=1}^{T} \exp(-f(\mathbf{x}_t^{\phi})) = \exp\left(-\sum_{t=1}^{T} f(\mathbf{x}_t^{\phi}) - \frac{1}{2\lambda}\|\phi\|_2^2\right). \tag{3}$$

**Training.** Maximum a posteriori (MAP) estimation of the optimizer parameters leads to a degenerate distribution and point estimation without uncertainty. Moreover, directly maximizing the optimizer posterior in Eq. 3 via Markov chain Monte Carlo (MCMC) would encounter computational intractability considering the high-dimensional space of the optimizer space parameterized in L2O by neural network weights. Instead, we introduce a variational distribution $q(\phi; \boldsymbol{\theta})$ (Kingma & Welling, 2013) with the variational parameter $\boldsymbol{\theta}$ and accordingly we introduce the following objective function whose minimization is equivalent to maximizing the evidence lower bound (ELBO) (see Appendix D for the derivation):

$$-\text{KL}[q(\phi; \boldsymbol{\theta})||p(\phi|\mathbf{z}_T)] = -\mathbb{E}_{\phi \sim q(\phi; \boldsymbol{\theta})} \sum_{t=1}^{T} f(\mathbf{x}_t^{\phi}) - \text{KL}[q(\phi; \boldsymbol{\theta})||p(\phi)]. \tag{4}$$

We refer to the L2O under the new meta-training loss (4) w.r.t $\boldsymbol{\theta}$ as uncertainty-aware L2O (**UA-L2O**), which can be trained end-to-end with the Gaussian parametrization and the reparametrization

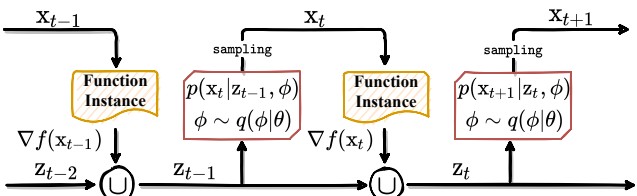

**Figure 1:** Feed-forward computational chart for training UA-L2O. Notations are as in the main text.

trick (Kingma & Welling, 2013; Higgins et al., 2016). The feed-forward computational chart for training UA-L2O is depicted in Figure 1.

**Deployment**. With the well-trained UA-L2O and the found $\boldsymbol{\theta}^*$, we ideally assume the function values of the trajectory are submartingales that $\mathbb{E}_{\boldsymbol{\phi}\sim q(\boldsymbol{\phi};\boldsymbol{\theta}^*)}f(\mathbf{x}_{t+1}^{\boldsymbol{\phi}}) \leq f(\mathbf{x}_t), \mathbb{E}_{\boldsymbol{\phi}\sim q(\boldsymbol{\phi};\boldsymbol{\theta}^*)}|f(\mathbf{x}_t^{\boldsymbol{\phi}})| < \infty$. Then based upon martingale convergence theorem (Doob, 1953) $\lim_{t\to\infty} f(\mathbf{x}_t) = f(\mathbf{x}_\infty)$ exists almost surely, i.e. it converges with infinite optimization iterations. Within the finite steps, UA-L2O has a well-defined approximated posterior $q(\boldsymbol{\phi};\boldsymbol{\theta}^*)$, and hence the baked-in ability to quantify optimizer uncertainty at the solution. Therefore, the posterior of solutions is calculated by the integral with the variational distribution $q(\boldsymbol{\phi};\boldsymbol{\theta}^*)$ as:

$$p(\mathbf{x}^*|\mathbf{z}_T) = \int p(\mathbf{x}^*|\mathbf{z}_T,\boldsymbol{\phi})q(\boldsymbol{\phi};\boldsymbol{\theta}^*)d\boldsymbol{\phi}, \tag{5}$$

which can be estimated by Monte Carlo sampling. Specifically, for estimating the solution posterior, the learned model is run 10,000 times with random initialization and different trajectories $\mathbf{z}_t$ accordingly. The optimizer parameter $\boldsymbol{\theta}$ is sampled from $q(\boldsymbol{\theta}|\boldsymbol{\phi}^*)$ once per iteration, following variational inference as in Kingma & Welling (2013) and Higgins et al. (2016).

## 4 EXPERIMENTS

We examine our proposed UA-L2O on optimizing: (i) non-convex test functions, (ii) loss functions in data privacy attack, and (iii) energy functions in predicting 3D protein-protein interactions (protein docking). We compare UA-L2O with three non-Bayesian methods: manually-designed Adam (Kingma & Ba, 2014) and particle swarm optimization (PSO) (Kennedy & Eberhart, 1995) as well as DM_LSTM (Andrychowicz et al., 2016a), an L2O method. We also compare to Adam and PSO with hyper-parameter Bayesian optimization (Adam-BO and PSO-BO), Adam with learning rate ensembles (Adam_lr_Ensemble) and Adam with stochastic gradient MCMC (Adam_Noisy_Gradient). Lastly we compare to Bayesian active learning (BAL) (Cao & Shen, 2020).

For Adam and PSO, we make their hyper-parameters probabilistic to calculate the posterior as shown in Table 6 in Appendix C. All algorithms are run for 10,000 times with random initialization to obtain the empirical posterior distributions $p(\mathbf{x}^*|\mathbf{z}_T)$, following Eq. 5 for UA-L2O and its counterparts for others. We access the optimization uncertainty with the following expression (Ortega et al., 2012; Cao & Shen, 2020):

$$P\left(\mathrm{lb}_\sigma \leq \|\mathbf{x}^* - \hat{\mathbf{x}}\|_2 \leq \mathrm{ub}_\sigma\right) = 1 - \sigma, \tag{6}$$

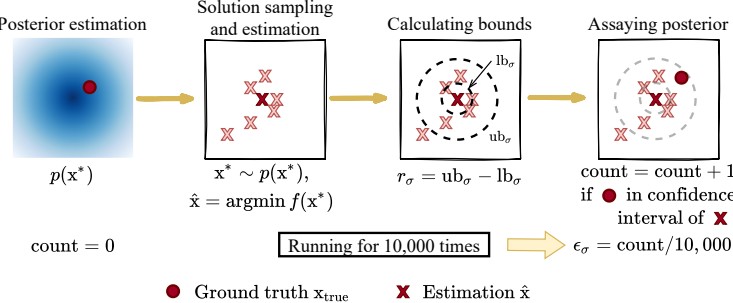

**Figure 2:** Computational procedure for uncertainty quantification: given a designated confidence score $\sigma$, calculating the length of confidence interval $r_\sigma$ and its corresponding estimated confidence $\epsilon_\sigma$.

where $lb_\sigma$ and $ub_\sigma$ are the lower and upper bounds at the confidence level of $\sigma$, and we choose the one with the lowest function value (we also study the mean value in Appendix E) to be the estimated solution $\hat{x}$. Ablation on different runs and comparison with competitors in wall-clock time are presented in Appendix E. For the choice of assessing the proximity of solutions rather than function values $|f(x^*) - f(\hat{x})|$, the reason is that in many optimization applications (e.g. privacy attack and protein docking), the objective function actually plays the role of the "tool" instead of the "goal", thus the function value is not the key metric to assess the quality of solutions in particular when the objective function is non-convex and noisy.

**Evaluation metrics.** UA-L2O is designed for uncertainty awareness in the optimizer. To assess the calibration of uncertainty quantification (UQ), we use the accuracy of confidence, following (Ortega et al., 2012; Cao & Shen, 2020). In other words, we compare expected confidence levels $\sigma$ as in Eq. 6 (0.8 and 0.9 in this paper) and the empirical probability $\varepsilon_\sigma$ of "success" when the corresponding confidence intervals indeed contain the global optima. When there is a tie in the major assessment metric $\sigma - \varepsilon_\sigma$, the tightness of the confidence interval for the solution's proximity to the global optimum (Eq. 6), $r_\sigma = ub_\sigma - lb_\sigma$, serves as a tie-breaker. A schematic illustration of the uncertainty assessment is shown in Figure 2.

Although UA-L2O is designed for the uncertainty awareness rather than the quality of optimization solutions, we still assess potential benefit in optimization performance, using the expected distance between the optimized solutions and the global optimum $\mathbb{E}_{x^* \sim p(x^*|z_T)} \|x^* - x_{\text{true}}\|_2$. The lower distances indicate the better solution quality (optimization performance).

**Two-stage training.** Due to the extremely high-dimensional optimizer space and the rugged posterior landscape, it is over-challenging to directly train the model through the equivalency of ELBO optimization in Equation (4). We therefore perform two-stage training where we first train our model in a non-Bayesian way and then use the resulting $\theta$ as the warm start for fine-tuning the mean variational parameters in the second stage. We examine the performance of the warm-start model in Appendix E to show the benefit of the second-stage training.

**Implementations.** The model is implemented in Tensorflow 1.13 (Abadi et al., 2016) and optimized by Adam (Kingma & Ba, 2014). For the L2O architecture, we use the coordinate-wise LSTM from (Andrychowicz et al., 2016a) containing 10,282 free parameters. The meta-training and test data are described in Appendix B. For all experiments, the length of LSTM is set to be 20, with 5,000 training epochs for both training stages. We tune the $\lambda$ values in equations (3) and (4) from $\{0.1, 1, 10\}$, validated by validation performance in test functions and by the tightness of confidence intervals in privacy attack and protein docking. The ablation study on $\lambda$ is presented in Appendix C showing that UA-L2O is not sensitive to the value of $\lambda$.

## 4.1 NON-CONVEX TEST FUNCTIONS

**Setup.** We first evaluate the performance on the test functions in the global optimization benchmark (Jamil & Yang, 2013). We choose three extremely rugged, non-convex functions, Rastrigin, Ackley and Griewank with analytical forms shown in Appendix B, paired with dimensionalities in $\{3, 6, ..., 30\}$.

**Results.** We compare UA-L2O with competitors in optimization and UQ performances shown in Figure 3. In the most relevant assessment metric, the accuracy of confidence levels (bottom two rows), UA-L2O performs the best in all but few cases for all the three functions, outperforming hand-engineered optimizers, non-Bayesian or Bayesian, and L2O without uncertainty awareness (DM_LSTM). The few exceptions, such as the Griewank function in 27 dimensions, could not be attributed to the higher dimensionality (as that was not the case for UA-L2O for Rastrigin and Ackley functions). DM_LSTM also showed overconfidence in such exceptions but the few exceptions cannot be attributed to L2O either.

Given that UA-L2O consistently outperforms competitors in confidence accuracy, comparing the tightness of confidence intervals alone (Figure 9) is not meaningful. Note that other methods, when having tighter confidence intervals, were often too overconfident to realize that the true global optima were inside the intervals much less than their confidence levels suggested.

We also examined potential but not intended benefits of UA-L2O in optimization performance (top row of Figure 3). Interestingly, UA-L2O even improved optimization performances for Ackley and Griewank functions when the dimensionality is above 12, while being slightly worse in Rastrigin.

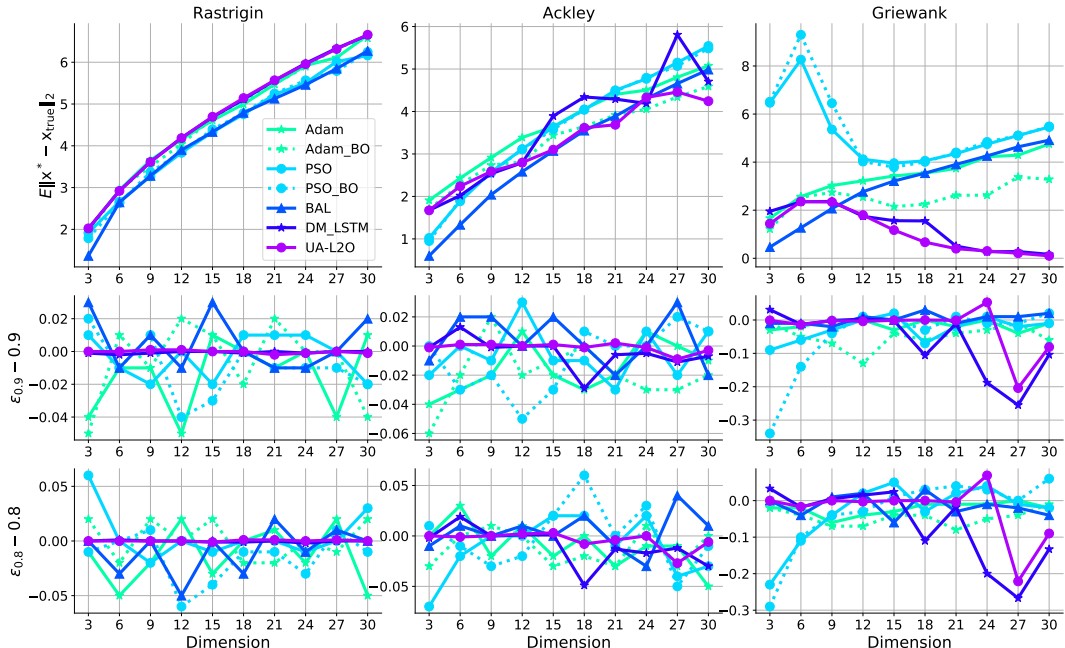

**Figure 3:** Optimization and uncertainty performance of different methods in three non-convex test functions. Different column represents different functions, and each row stands for: (i) 1st row is for the non-intended optimization performance, the lower the better; (ii) 2nd & 3rd rows, the most important metric for the intended uncertainty calibration, are for the precision of the estimated confidence, lower values indicating more accurate posterior estimation. The corresponding confidence intervals are shown in Appendix C.

These results echo our conjecture that, although our central goal is not for better optimization solutions but for their better uncertainty awareness, better uncertainty calibration could sometimes improve optimization performances.

We further plot the confidence for solution regions segmented in $\|\mathbf{x}^* - \hat{\mathbf{x}}\|_2$ versus the corresponding optimization performance $\|\mathbf{x}^* - \mathbf{x}_{\text{true}}\|_2$ (see Appendix A for detailed procedure) in Figure 10. We find that when UA-L2O achieves better optimization performance (Ackley and Griewank as opposed to Rastrigin), it generates solutions with higher confidence in the regions closer to the global optima and with narrower distributions (more certain). In contrast the competitors do not show such trends.

## 4.2 DATA PRIVACY ATTACK

**Setup.** We next apply our model to an application that critically needs UQ. As many machine learning models are deployed publicly, it is important to avoid leaking private and sensitive information, such as financial data and health data. Data privacy attack (Nasr et al., 2018) studies this problem by playing the role of hackers and attacking machine learning models to quantify the risk of privacy leakage. Better attacks would help models to be better prepared for privacy defense.

We use the model and dataset in (Cao et al., 2019b), where each input has 9 features involving patient genetic information shown in Appendix B, and the output is the probability of the clinical significance (pathogenicity) for genetic variants in patients. We study the following model inversion attack (Fredrikson et al., 2015): given 5 features $\mathbf{x}' \in [0, 1]^5$ out of 9 and the label $y$ of each patient, the privacy attack aims to recover the rest 4 features $\mathbf{x}_{\text{true}} \in [0, 1]^4$ (potentially sensitive patient information). Therefore, the optimization is $\min_{\mathbf{x} \in [0,1]^4} \sum (g(\mathbf{x}', \mathbf{x}) - y)^2$ for all patients where $g$ is a trained predictor. The closeness between the optimized and real input features can quantify the risk of information leakage and the quality of the attack. We evaluate our method for all test cases in (Cao et al., 2019b).

**Results.** We report the UQ and optimization performances of UA-L2O in Table 1. In UQ, UA-L2O's performance (the accuracy of estimated confidence) stands out against competitors, although not as dominant as it does to competitors in test functions. Even in optimization, UA-L2O outperforms all competitors, which again demonstrate the potential benefit of uncertainty-awareness to optimization.

**Table 1:** Optimization and uncertainty performance of different methods in genetic data privacy attack.

| Method | $\mathbb{E}\|\mathbf{x}^* - \mathbf{x}_{\text{true}}\|_2$ | $|\epsilon_{0.9} - 0.9|$ | $|\epsilon_{0.8} - 0.8|$ | $r_{0.9}$ | $r_{0.8}$ |
|---|---|---|---|---|---|
| Adam | 0.39 | 0.90 | 0.80 | 0.08 | 0.06 |
| Adam_BO | 0.38 | 0.90 | 0.80 | 0.08 | 0.05 |
| Adam_lr_Ensemble | 0.35 | 0.90 | 0.80 | 0.01 | 0.009 |
| Adam_Noisy_Gradient | 0.57 | 0.90 | 0.80 | 0.01 | 0.01 |
| PSO | 0.54 | 0.08 | 0.10 | 0.54 | 0.34 |
| PSO_BO | 0.53 | 0.06 | 0.15 | 0.48 | 0.42 |
| BAL | 0.52 | 0.90 | 0.80 | 0.01 | 0.01 |
| DM_LSTM | 0.34 | 0.09 | 0.77 | 0.09 | 0.02 |
| UA-L2O | **0.30** | 0.25 | 0.27 | 0.06 | 0.04 |

Note that PSO had slightly more accurate confidence but much looser intervals, suggesting flat posteriors and leading to worse optimization.

We argue that, unlike pure optimization problems in test functions, the optimization for privacy attack involves the generalization issue in the machine learning paradigm: less generalizable samples pose challenges to not only optimization but also UQ. To verify this argument, we plot optimization performance for individual samples versus their estimated posterior precision in Figure 4 showing a positive correlation: samples easier to generalize are closer to the optima (more leftward in the $x$-axis), and more precise in estimated confidence (more downward in the $y$-axis). The tightness of confidence intervals is also well correlated with solution quality.

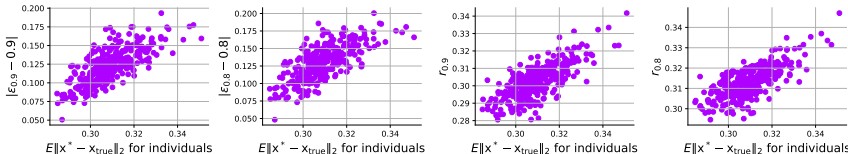

**Figure 4:** Optimization performance versus UQ results ($\epsilon_{0.9}, \epsilon_{0.8}, r_{0.9}$ and $r_{0.8}$) of UA-L2O for 318 test samples in data privacy attack.

Similar to that in Section 4.1, we also plot confidence versus optimization performance in Figure 11, to further demonstrate the calibration capability of UA-L2O. To demonstrate the practical use of developed UQ, we conduct an experiment on "failure" case detection, that we use assessed uncertainties to probe the optimization performance of individual runs. Results in Appendix F show UA-L2O provides the best detection.

### 4.3 Energy Functions for Protein Docking

**Setup.** We lastly examine UA-L2O using a bioinformatics application: predicting the 3D structures of protein-complexes (Smith & Sternberg, 2002) or protein docking. *Ab initio* protein docking can be recast as optimizing a noisy and expensive energy function in a high-dimensional conformational space (Cao & Shen, 2020). While solving such optimization problems still remains difficult, quantifying the uncertainty of resulting solutions is even more challenging. The compared state-of-the-art method is BAL (Cao & Shen, 2020).

We calculate the energy function (objective function $f(\mathbf{x})$) in a CHARMM19 force field as in (Moal & Bates, 2010). 25 protein-protein complexes are chosen from the protein docking benchmark set 4.0 (Hwang et al., 2010) as the training set, which is shown in Appendix B. For

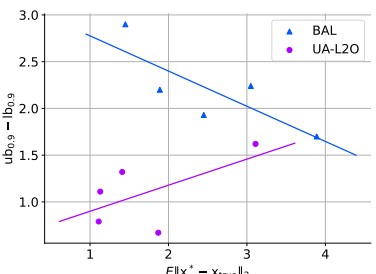

**Figure 5:** Scatter plot of optimization performance versus the length of confidence interval for five cases in protein docking. The Pearson correlation coefficient achieved by BAL is -0.80, and that by UA-L2O is **0.59**.

each complex, we choose 5 starting points (top-5 structure predictions from ZDOCK (Pierce et al., 2014)). In total, our training set includes 125 samples. Moreover, we parameterize the search space for flexible docking as $\mathbb{R}^{12}$ as in BAL. The resulting $f(\mathbf{x})$ is fully differentiable in the search space. We only consider 100 interface atoms due to the computational concern. The number of iterations for one training epoch is 600 and in total we have 5,000 training epochs. Both BAL and UA-L2O have 600 iterations during the testing stage.

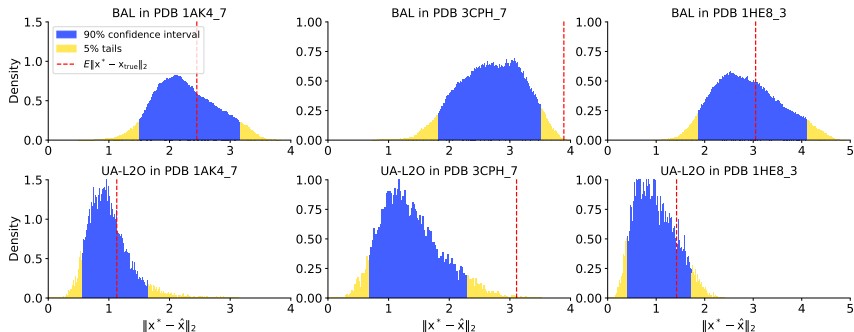

**Figure 6:** Estimated posterior distributions, confidence intervals and ground truth solutions for cases 1AK4_7, 3CPH_7 and 1HE8_3 in protein docking.

**Table 2:** Optimization and uncertainty performance of BAL and UA-L2O in protein docking.

| Target PDB model (docking difficulty) | $\mathbb{E}\|\mathbf{x}^* - \mathbf{x}_{\text{true}}\|_2$ (Å) | | $\mathbb{E}\|\mathbf{x}^* - \mathbf{x}_{\text{true}}\|_2 \in [\text{lb}_{0.9}, \text{ub}_{0.9}]$? | | $\text{ub}_{0.9} - \text{lb}_{0.9}$ (Å) | |
|---|---|---|---|---|---|---|
| | **BAL** | **UA-L2O** | **BAL** | **UA-L2O** | **BAL** | **UA-L2O** |
| 1AHW_3 (easy) | 1.89 | **1.11** | No | Yes | 2.20 | 0.79 |
| 1AK4_7 (easy) | 2.45 | **1.13** | Yes | Yes | 1.93 | 1.11 |
| 3CPH_7 (medium) | 3.89 | **3.11** | No | No | 1.70 | 1.62 |
| 1HE8_3 (medium) | 3.05 | **1.42** | Yes | Yes | 2.24 | 1.32 |
| 1JMO_4 (difficult) | **1.45** | 1.87 | No | No | 2.90 | 0.67 |

**Results.** We compare UA-L2O and BAL in Table 2. For UQ, UA-L2O not only achieves a more accurate confidence as shown in Figure 6 and Appendix C (three of five cases had ground truths located in 90% confidence intervals), but also has a clearer trend that the better solutions correspond to tighter intervals (more certain) as in Figure 5 (with Pearson correlation coefficient of 0.59 versus -0.8 for BAL). Even for optimization, UA-L2O outperforms other methods in four out of five cases. The results show significant advantages of UA-L2O compared with the state-of-the-art BAL in both UQ and optimization for protein docking.

## 4.4 RESULT SUMMARY

We briefly summarize the results as follows.

- In general, UA-L2O achieves better **optimization performance** with a few exceptions. This echos the hypothesis that the assessment of uncertainty is able to enhance the search efficiency and effectiveness during optimization.
- UA-L2O provides the most accurate **posterior estimation** most of the time. This empirical evidence verifies the effectiveness of our uncertainty-aware L2O inspired from theory.
- Accurate posterior estimation is a key for UA-L2O's superior **calibration capability**, such that UA-L2O generates solutions closer to optima with higher confidence and tighter confidence intervals, whereas competitors suffer from overconfidence or miscalibration. Such calibration capability is critical to detect the failure cases in real-world applications.

## 5 CONCLUSIONS

Current optimization algorithms, even with uncertainty-awareness, do not address the uncertainty arising within the optimizer itself. To close the gap, in this paper we attempt to ask and address three fundamental questions, why modelling the optimizer uncertainty, what defines the optimizer uncertainty, and how to enabling UQ during optimization. We first emphasize the uncertainty arising from the optimizer is a crucial source that directly responses for the end solutions deriving, except for data- and model-uncertainty. Next, we define the prior and likelihood of the optimizer which determines the optimizer posterior and the optimizer uncertainty. We further leverage learning to optimize (L2O) for the optimizer parameterization, treating an optimizer as a random sample from an algorithmic space of iterative optimizers, with the end-to-end training pipeline built via variational inference. Extensive experiments show UA-L2O achieves the optimization performance, superior in two out of three test functions with the high variable dimension, best in the loss function in data privacy attack, and exceeded in the energy function for protein docking in four out of five cases. Besides, it delivers the most accurate posterior estimation and calibration capability, with the solutions closer to the ground truth of a larger population and tighter confidence intervals. Our study represents the first effort to recognize and quantify the uncertainty of the optimization algorithm, with extensive support from numerical results and analysis.

ACKNOWLEDGMENT

This project was in part supported by NSF (CCF-1943008 to YS, CCSS-2113904 to ZW). Portions of this research were conducted with the advanced computing resources provided by Texas A&M High Performance Research Computing.

REPRODUCIBILITY STATEMENT

To ensure reproducibility, we use the Machine Learning Reproducibility Checklist v2.0, Apr. 7 2020 (Pineau et al., 2021).

- For all **models** and **algorithms** presented,
    - **A clear description of the mathematical settings, algorithm, and/or model.** We clearly describe all of the settings, formulations, and algorithms in Section 3.
    - **A clear explanation of any assumptions.** We state our assumptions clearly in Section 3.
    - **An analysis of the complexity (time, space, sample size) of any algorithm.** We do not make the analysis.
- For any **theoretical claim**,
    - **A clear statement of the claim.** We provide a clear statement in Section 3.
    - **A complete proof of the claim.** We do not make theoretical proofs.
- For all **datasets** used, check if you include:
    - **The relevant statistics, such as number of examples.** We provide all the related statistics in Section 4 and Appendix B.
    - **The details of train/validation/test splits.** We give this information in our repository.
    - **An explanation of any data that were excluded, and all pre-processing step.** We give this information in our repository.
    - **A link to a downloadable version of the dataset or simulation environment.** Our repository contains all of the instructions to run experiments on the datasets in this work. We put it in supplementary materials.
    - **For new data collected, a complete description of the data collection process, such as instructions to annotators and methods for quality control.** We do not collect or release new datasets.
- For all shared **code** related to this work, check if you include:
    - **Specification of dependencies.** We give installation instructions in the README of our repository.
    - **Training code.** The training code is available in our repository.
    - **Evaluation code.** The evaluation code is available in our repository.
    - **(Pre-)trained model(s).** We do not release trained models.
    - **README file includes table of results accompanied by precise command to run to produce those results.** We include a README with detailed instructions to reproduce all of our experimental results.
- For all reported **experimental results**, check if you include:
    - **The range of hyper-parameters considered, method to select the best hyper-parameter configuration, and specification of all hyper-parameters used to generate results.** We provide all details of hyper-parameter tuning in Section 4.
    - **The exact number of training and evaluation runs.** Ten thousand independent repetitions are conducted for each experiments.
    - **A clear definition of the specific measure or statistics used to report results.** We provide all details of evaluation metrics in Section 4.
    - **A description of results with central tendency (e.g. mean) & variation (e.g. error bars).** We report confidence intervals for all experiments in Section 4.

- **The average runtime for each result, or estimated energy cost.** We do not report the running time or energy cost.
- **A description of the computing infrastructure used.** A clear description is presented in Section 3.

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

## Appendix

## A  Confidence Estimation Procedures

In evaluation, for the confidence versus optimization performance scatter plot, we first perform Monte Carlo sampling on the solution posterior, and then bin the solutions to estimate their fraction and corresponding average distance to the optimum. Detailed procedures can be found in Algorithms 1.

---

**Algorithm 1:** Confidence estimation with corresponding optimization performance

**Input:** Sampled solutions $\mathbf{x}^*$ with size $N$, estimated solution $\hat{\mathbf{x}}$, known optimum $\mathbf{x}_{\text{true}}$, bin width $w$.
**Initialize:** $S = \{\}$.
1. Calculate $\|\mathbf{x}^* - \hat{\mathbf{x}}\|_2$ for all sampled solutions, and bin them with $w$.
**for** each bin $b$ **do**
 2. Calculate the population fraction $|b|/N$.
 3. Calculate average distance to $\mathbf{x}_{\text{true}}$: $\text{Mean}(\{\|\mathbf{x}^* - \mathbf{x}_{\text{true}}\|_2 | \mathbf{x}^* \in b\})$.
 4. $S \leftarrow S \cup (|b|/N, \text{Mean}(\{\|\mathbf{x}^* - \mathbf{x}_{\text{true}}\|_2 | \mathbf{x}^* \in b\}))$.
**end for**
**Return:** Scatter point set $S$.

---

## B  Data Description

The analytical forms of three test functions are shown in Table 3. The meta-training data are a broad family of similar functions (Cao et al., 2019a). For Rastrigin, it is $f_D(\mathbf{x}) = \|A\mathbf{x} - b\|_2^2 - c\sum_{i=1}^{D} 10\cos(2\pi x_i) + 10D$ where $A, b, c$ are sampled from i.i.d normal distributions; for Ackley, it is $f_D(\mathbf{x}) = -20\exp(-0.2\sqrt{0.5\|A\mathbf{x} - b\|_2^2}) - c\sum_{i=1}^{D}\exp(\cos(2\pi x_i)/D) + \exp(1) + 20$; for Griewank, it is $f_D(\mathbf{x}) = 1 + \sum_{i=1}^{D}\|A\mathbf{x} - b\|_2^2/4000 - c\prod_{i=1}^{D}\cos(x_i)$.

For privacy attack, we use 9-dimension features of privacy attack data generated from MutPred2 (Pejaver et al., 2017), which captures mutational impacts on protein structure, dynamics, and function, and grouped hierarchically into a custom oncology based on their inherent relationships. The list of features can be found in Table 4. The meta-training is performed on 1,593 patients and we evaluate in 318 patients (Cao et al., 2019b).

For protein docking, the PDB IDs of meta-training samples are shown in Table 5 and we test in proteins with PDB in Table 2.

**Table 3:** The analytical forms of test functions $f_D(\mathbf{x})$ with the dimension $D$, and their corresponding flatness around global optima, estimated with the norm of Hessain matrix.

| Function | Analytic form |
|---|---|
| Rastrigin | $f_D(\mathbf{x}) = \|\mathbf{x}\|_2^2 - \sum_{i=1}^{D} 10\cos(2\pi x_i) + 10D$ |
| Ackley | $f_D(\mathbf{x}) = -20\exp(-0.2\sqrt{0.5\|\mathbf{x}\|_2^2}) - \sum_{i=1}^{D}\exp(\cos(2\pi x_i)/D) + \exp(1) + 20$ |
| Griewank | $f_D(\mathbf{x}) = 1 + \sum_{i=1}^{D}\|\mathbf{x}\|_2^2/4000 - \prod_{i=1}^{D}\cos(x_i)$ |

## C  More Evaluation Results

We provide more evaluation results as follows. (i) The ablation of $\lambda$ values on three test functions is shown in Figure 7. (ii) The UQ results for test cases 1AHW_3 and 1JMO_4 in protein docking are plotted in Figure 8. (iii) Confidence intervals corresponding to results in Figure 3.

**Table 4:** 9-dimension feature description for data privacy attack experiment.

| Feature index | Property/molecular impact |
|---|---|
| 1 | Relative solvent accessibility |
| 2 | Allosteric site |
| 3 | Catalytic site |
| 4 | Secondary structure |
| 5 | Stability and conformation flexibility |
| 6 | Special structural signatures |
| 7 | Macromolecular binding |
| 8 | Metal binding |
| 9 | PTM site |

**Table 5:** 4-letter ID of proteins used in protein docking training set.

| Difficulty level | Protein Data Bank (PDB) code |
|---|---|
| Rigid | 1N8O, 7CEI, 1DFJ, 1AVX, 1BVN, 1IQD, 1CGI, 1MAH, 1EZU, 1JPS, 1PPE, 1R0R, 2I25, 2B42, 1EAW, 2JEL, 1BJ1, 1KXQ, 1EWY |
| Medium | 1XQS, 1M10, 1IJK, 1GRN |
| Flexible | 1IBR, 1ATN |

**Figure 7:** Optimization and uncertainty performance of UA-L2O with different $\lambda$ in three non-convex test functions. Different row represents different functions, and each column stands for: (i) 1st column is optimization performance, lower the better; (ii) 2nd column is precision of the estimated confidence, lower denoting more accurate posterior estimation; (iii) 3rd is the length of the confidence interval: only when with similar precision of the estimated confidence would a tighter interval indicating more certain solutions.

**Table 6:** The optimizer distributions over hyper-parameters in Adam and PSO.

| Methods | Optimizer Distribution Settings |
|---|---|
| Adam | $\log_{10}(\text{lr}) \sim \text{U}[-2, -1]$, $\beta_1 \sim \text{U}[0.9, 1.0]$, $\beta_2 \sim \text{U}[0.999, 1.0]$ |
| PSO | $w \sim \text{U}[0.5, 1.5]$, $C1 \sim \text{U}[1.5, 2.5]$, $C2 \sim \text{U}[1.5, 2.5]$ |

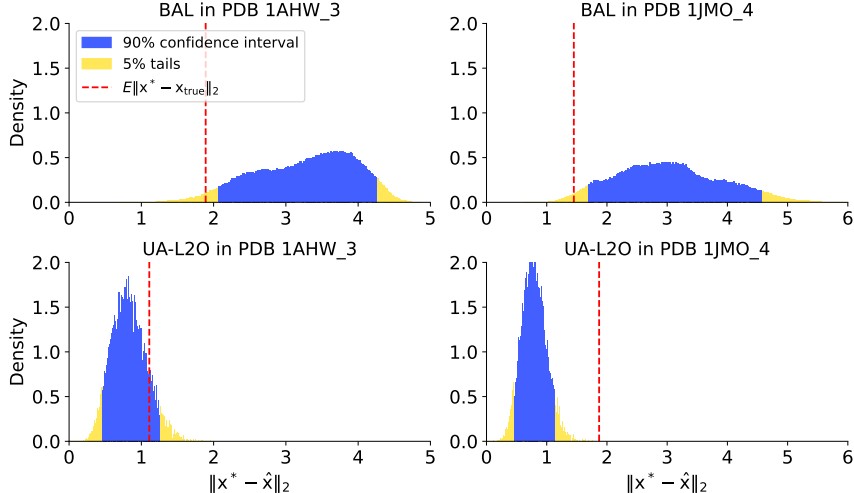

**Figure 8:** Estimated posterior distributions, confidence intervals and ground truth solutions for cases 1AHW_3 and 1JMO_4 in protein docking.

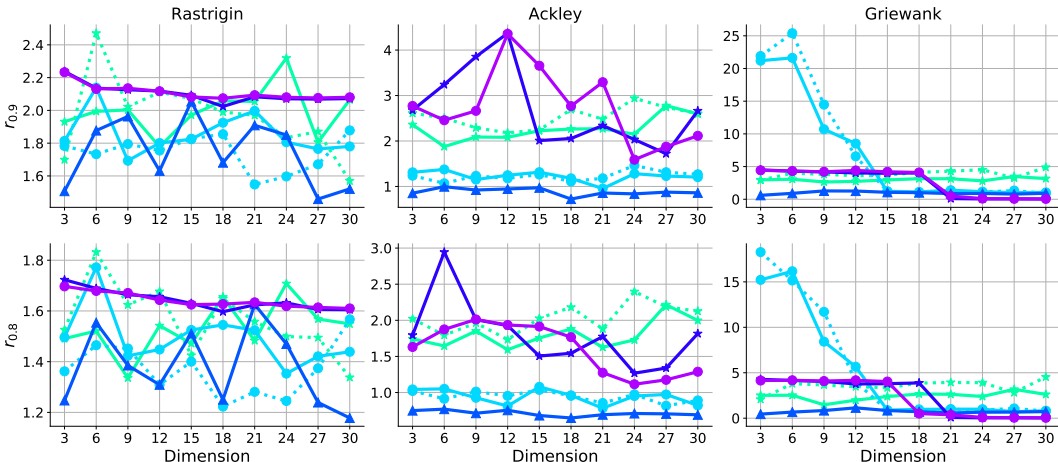

**Figure 9:** The corresponding confidence intervals to results in Figure 3, a secondary metric and tie-breaker for uncertainty calibration accuracy and a measure of the length of the confidence interval: only when the estimated confidence levels are similarly precise, tighter intervals indicates higher certainty about solutions

## D  COMPUTATION OF UA-L2O OBJECTIVE FUNCTION

We first rewrite the evidence as:

$$\log p(\mathbf{z}_T|\mathbf{z}_{t_0}) = \int q(\boldsymbol{\phi};\boldsymbol{\theta})\log p(\mathbf{z}_T|\mathbf{z}_{t_0})d\boldsymbol{\phi} = \int q(\boldsymbol{\phi};\boldsymbol{\theta})\log\frac{p(\mathbf{z}_T,\boldsymbol{\phi}|\mathbf{z}_{t_0})}{p(\boldsymbol{\phi}|\mathbf{z}_T)}d\boldsymbol{\phi}$$

$$= \int q(\boldsymbol{\phi};\boldsymbol{\theta})\log p(\mathbf{z}_T,\boldsymbol{\phi}|\mathbf{z}_{t_0})d\boldsymbol{\phi} - \int q(\boldsymbol{\phi};\boldsymbol{\theta})\log p(\boldsymbol{\phi}|\mathbf{z}_T)d\boldsymbol{\phi}$$

$$= -\int q(\boldsymbol{\phi};\boldsymbol{\theta})\log\frac{q(\boldsymbol{\phi};\boldsymbol{\theta})}{p(\mathbf{z}_T,\boldsymbol{\phi}|\mathbf{z}_{t_0})}d\boldsymbol{\phi} + \int q(\boldsymbol{\phi};\boldsymbol{\theta})\log\frac{q(\boldsymbol{\phi};\boldsymbol{\theta})}{p(\boldsymbol{\phi}|\mathbf{z}_T)}d\boldsymbol{\phi}$$

$$= -\text{KL}[q(\boldsymbol{\phi};\boldsymbol{\theta})||p(\mathbf{z}_T,\boldsymbol{\phi}|\mathbf{z}_{t_0})] + \text{KL}[q(\boldsymbol{\phi};\boldsymbol{\theta})||p(\boldsymbol{\phi}|\mathbf{z}_T)],$$

where $-\text{KL}[q(\boldsymbol{\phi};\boldsymbol{\theta})||p(\mathbf{z}_T,\boldsymbol{\phi}|\mathbf{z}_{t_0})]$ is ELBO and $\text{KL}[q(\boldsymbol{\phi};\boldsymbol{\theta})||p(\boldsymbol{\phi}|\mathbf{z}_T)]$ is our objective. Therefore, minimizing $\text{KL}[q(\boldsymbol{\phi};\boldsymbol{\theta})||p(\boldsymbol{\phi}|\mathbf{z}_T)]$ is equivalent to maximizing ELBO (Yin & Zhou, 2018).

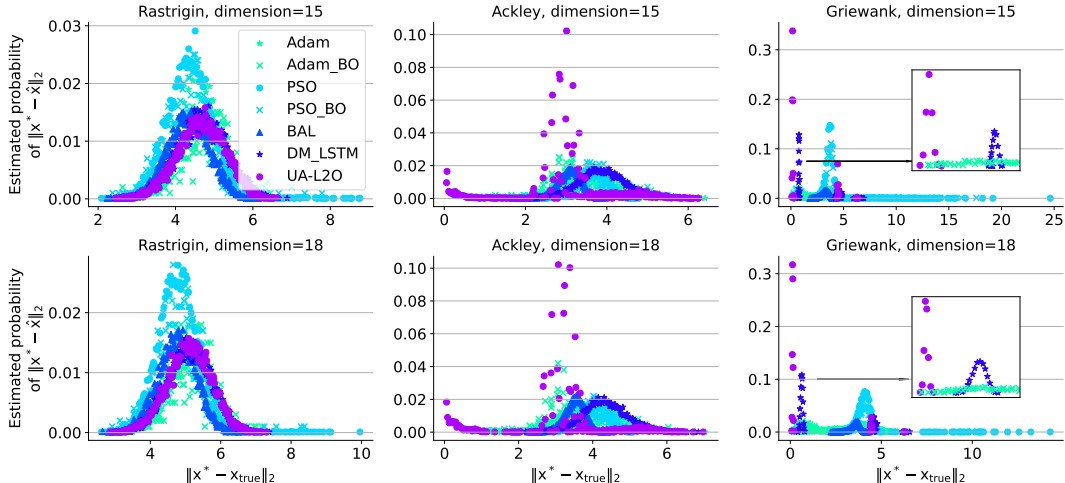

**Figure 10:** Scatter plot of confidence for solution regions segmented by $\|\mathbf{x}^* - \hat{\mathbf{x}}\|_2$, versus the corresponding optimization performance $\|\mathbf{x}^* - \mathbf{x}_{\text{true}}\|_2$ in three test functions. The peak closer to left side indicates the solution populations are in high confidence if they are closer to the optima.

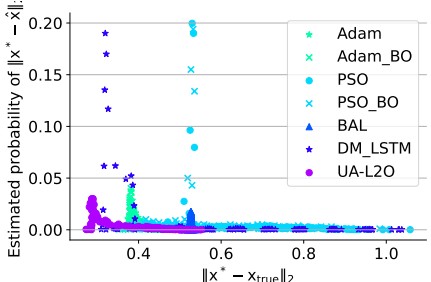

**Figure 11:** Scatter plot of confidence for solution regions segmented by $\|\mathbf{x}^* - \hat{\mathbf{x}}\|_2$, versus the corresponding optimization performance $\|\mathbf{x}^* - \mathbf{x}_{\text{true}}\|_2$ in data privacy attack.

We rewrite our objective as:

$$\mathrm{KL}[q(\boldsymbol{\phi};\boldsymbol{\theta})\|p(\boldsymbol{\phi}|\mathbf{z}_T)] = \mathbb{E}_{\boldsymbol{\phi}\sim q(\boldsymbol{\phi};\boldsymbol{\theta})}\Big\{\log q(\boldsymbol{\phi};\boldsymbol{\theta}) - \log p(\boldsymbol{\phi}|\mathbf{z}_T)\Big\}$$

$$= \mathbb{E}_{\boldsymbol{\phi}\sim q(\boldsymbol{\phi};\boldsymbol{\theta})}\Big\{\log q(\boldsymbol{\phi};\boldsymbol{\theta}) - \log p(\boldsymbol{\phi})p(\mathbf{z}_T|\mathbf{z}_{t_0},\boldsymbol{\phi})\Big\} + Z$$

$$= \mathbb{E}_{\boldsymbol{\phi}\sim q(\boldsymbol{\phi};\boldsymbol{\theta})}\Big\{\log \frac{q(\boldsymbol{\phi};\boldsymbol{\theta})}{p(\boldsymbol{\phi})} - \log p(\mathbf{z}_T|\mathbf{z}_{t_0},\boldsymbol{\phi})\Big\} + Z$$

$$= \mathbb{E}_{\boldsymbol{\phi}\sim q(\boldsymbol{\phi};\boldsymbol{\theta})} \sum_{t=t_0+1}^{T} f(\mathbf{x}_t^{\boldsymbol{\phi}}) + \mathrm{KL}[q(\boldsymbol{\phi};\boldsymbol{\theta})\|p(\boldsymbol{\phi})] + Z$$

## E  ABLATION STUDY

We perform ablation studies on sample size in uncertainty quantification of UA-L2O in Table 7, mean versus minimum of $\hat{\mathbf{x}}$ computation and wall-clock time comparison in Table 8, on privacy attack experiment on gene BRCA1/2 in Table 9. Results show UA-L2O is not sensitive to sample size or mean/minimum calculation of $\hat{\mathbf{x}}$, and it is comparably efficient with competitor optimizers.

Moreover, we include a study of the wall-time versus dimensionality ranging from 10 to 2,000, using the optimization of Rastrigin function in Figure 10. We report UA-L2O's wall-time for both meta-training and meta-testing. The results above show that the meta-training time grows almost linearly with the dimensionality when dimensionality >200 (which is partly attributed to the fact that the number of samples is now adopted as a value correlated with the dimension). So meta-training even

in millions of dimensions for deep-learning model parameters is expensive yet still manageable, especially using multiple GPU cores. Most importantly, meta-testing was much faster and the time remained flat with regards to the dimensions, that once UA-L2O is trained, its deployment remains fast and scales well to the dimensionality. In fact, L2O usually takes the setting of offline training (which could be time-consuming) and online deployment (which is fast enough and scales well) Chen et al. (2021).

We further compare with baselines as learning rate ensemble and stochastic gradient MCMC of Adam, shown in Table 11. We also make comparison with warm-start UA-L2O to show the improvement of UA-L2O training.

**Table 7:** Ablation of sample size in posterior estimation on BRCA1/2 privacy attack.

| Sample size | $\mathbb{E}\|\mathbf{x}^* - \mathbf{x}_{\text{true}}\|_2$ | $|\epsilon_{0.9} - 0.9|$ | $|\epsilon_{0.8} - 0.8|$ | $r_{0.9}$ | $r_{0.8}$ |
|---|---|---|---|---|---|
| 100 | 0.30 | 0.25 | 0.25 | 0.06 | 0.06 |
| 1,000 | 0.30 | 0.27 | 0.31 | 0.06 | 0.04 |
| 10,000 | 0.30 | 0.25 | 0.27 | 0.06 | 0.04 |

**Table 8:** Ablation of mean/minimum of $\hat{\mathbf{x}}$ computation in posterior estimation on BRCA1/2 privacy attack.

| | $\mathbb{E}\|\mathbf{x}^* - \mathbf{x}_{\text{true}}\|_2$ | $|\epsilon_{0.9} - 0.9|$ | $|\epsilon_{0.8} - 0.8|$ | $r_{0.9}$ | $r_{0.8}$ |
|---|---|---|---|---|---|
| Mean | 0.31 | 0.27 | 0.26 | 0.07 | 0.04 |
| Min | 0.30 | 0.25 | 0.27 | 0.06 | 0.04 |

**Table 9:** Comparison of wall-clock time on BRCA1/2 privacy attack.

| | Adam | Adam_BO | PSO | PSO_BO | BAL | DM_LSTM | UA-L2O |
|---|---|---|---|---|---|---|---|
| Wall-Clock Time (s) | 30 | 33 | 22 | 24 | 43 | 35 | 41 |

**Table 10:** Comparison of wall-clock time on Rastrigin function with different dimensionality.

| Dimension | Meta-training (s) | Meta-test (s) |
|---|---|---|
| 10 | 607 | 36 |
| 100 | 610 | 38 |
| 200 | 662 | 37 |
| 500 | 1,059 | 38 |
| 1,000 | 1,998 | 38 |
| 2,000 | 3,904 | 38 |

**Table 11:** More baseline comparison of UA-L2O on privacy attack of gene BRCA1/2.

| | $\mathbb{E}\|\mathbf{x}^* - \mathbf{x}_{\text{true}}\|_2$ | $|\epsilon_{0.9} - 0.9|$ | $|\epsilon_{0.8} - 0.8|$ | $r_{0.9}$ | $r_{0.8}$ |
|---|---|---|---|---|---|
| UA-L2O Warm-Start Only | 0.76 | 0.55 | 0.57 | 0.40 | 0.30 |
| UA-L2O | 0.30 | 0.25 | 0.27 | 0.06 | 0.04 |

# F   UTILIZATION OF OPTIMIZER UNCERTAINTY

We demonstrate the "usefulness" of the proposed optimizer uncertainty via performing the experiment of "failure detection" as a basic downstream task.

Specifically, an optimization solution is defined as "failure" if its distance to the global optimum is beyond a threshold $0.5\sqrt{d}$ (where $d$ is the dimension and the threshold was chosen for approximately balanced success/failure cases). Instead of fixing the confidence level $\sigma$ and estimating the confidence interval $r$ (as observed in Figure 4 that better optimization correlated better with tighter confidence intervals), here we let each method estimate the confidence level $\sigma$ at fixed $r < 0.5\sqrt{d}$ ("success") for the optimization solution in each trajectory and use $\sigma$ to classify the quality of such solutions over 1000 trajectories. The classification performance was assessed using the area under

the receiver-operator characteristic curve (AUROC) and the area under the precision-recall curve (AUPRC). The results on privacy attack are shown in Table 12, indicating the best posterior estimation of UA-L2O.

Table 12: "Failure" case detection with optimizer uncertainties on privacy attack.

|  | Adam | Adam_BO | PSO | PSO_BO | BAL | DM_LSTM | UA-L2O |
|---|---|---|---|---|---|---|---|
| AUROC | 0.53 | 0.59 | 0.74 | 0.78 | 0.00 | 0.37 | 0.99 |
| AUPRC | 0.97 | 0.98 | 0.73 | 0.90 | 0.00 | 0.95 | 0.99 |

