# OpenReview forum: "Bayesian Modeling and Uncertainty Quantification for Learning to Optimize: What, Why, and How"
_ICLR.cc/2022/Conference — ICLR 2022 Poster_

### Official Review · Reviewer_Q3k1 · 2021-10-19

**Correctness:** 3
**Technical Novelty And Significance:** 3
**Empirical Novelty And Significance:** 3
**Recommendation:** 6
**Confidence:** 4

**Main Review:**

Treating optimizers as random elements in an algorithmic space is a novel concept to the best of my knowledge. Endowing priors on updating rules and defining likelihoods on the iterates also seem to be new in the Bayesian nonparametrics literature. In addition to data and model uncertainties, this paper brought attention to the role and importance of modeling optimizer uncertainty.

Here are some comments / questions:
1. It is known that the variational distribution often underestimates uncertainty, e.g., see ``Covariances, Robustness, and Variational Bayes" by Giordano et al. (2018). However from Figure 3, it seems that UA-L2O is still able to capture the estimated confidence quite well for low dimensions. So how did UA-L2O solve this underestimation problem?

2. Is the neural network architecture of $g$ fixed? In the experiments, it seems to me that the authors modeled $g$ using LSTM. However this contradicts with Definition 1 because $g$ is allowed to change according to the prior?

3. Why $\hat{\mathbf{x}}$ is not computed using mean of the posterior distribution of the solutions $\mathbf{x}^*$? Is there a reason to use the minimum of the generated solutions and not the posterior mean?

4. UA-L2O seems to have trouble optimizing Rastrigin, as evident by the plot of $E||x^*-x_{\text{true}}||_2$ against dimension $D$ in Figure 3, which looks like it is increasing to infinity. However the structure of Rastrigin seems to be the simplest among all three test functions. Can the authors give some intuition behind this?

5. For Ackley and Griewank in Figure 3, is there a reason why there is a dip in dimension $27$ in the precision of the estimated confidence (2nd column)?

6. In (3), what is $\mathbf{x}_t^{\phi}$?

7. There are quite a number of grammar mistakes and some parts are quite puzzling. For example, the line after (2), Prior arts on hyper-parameter?

TYPOS:
1. The first line in Section 3.1, am -> an
2. In (3), $\exp(-\lambda\|\boldsymbol{\phi}\|_2^2)$ -> $\exp(-\frac{1}{2\lambda}\|\boldsymbol{\phi}\|_2^2)$

**Summary Of The Paper:**

The authors looked at uncertainty quantification (UQ) of optimization algorithms. Working under the L2O framework and adopting the Bayesian approach, they used a neural network to model the updating rule, and endowed the network weights with a prior. By treating the solution iterates and their gradients as data, the authors defined a conditional likelihood on them. The resulting posterior on the network weights is then approximated using a variational algorithm. To conduct inference, the authors computed the predictive density of the solution through Monte Carlo sampling. This end-to-end pipeline is called uncertainty aware L2O (UA-L2O) and it is applied to quantify uncertainty in various learning tasks.

**Summary Of The Review:**

I think UA-L2O is a novel and promising method that might lead to further research in quantifying optimizer's uncertainty.

---

> ### Author Response · Authors · 2021-11-22
> **Response to reviewer Q3k1**
>
> We thank the reviewer for the interest in our manuscript and the thoughtful comments.
>
> 1. *It is known that the variational distribution often underestimates uncertainty*
>
> The reviewer raised an excellent point that caught our attention.  Thank you for the reference.  We don’t think that UA-L2O “solved” the underestimation issue, at least it wasn’t designed to.  A possibly related observation to echo the reviewer’s comment is that the confidence levels were relatively less tight in lower dimensions for two of the three functions, suggesting possibly more flat posteriors.  We appreciate the reference and would look into more Bayesian posterior inference techniques including the suggested for future improvement.
>
> 2. *Neural network architecture and the prior*
>
> The adopted L2O architecture (LSTM) is fixed while the parameters can be varying and learned. So the prior is allowed to change in the space of NN parameters, which conforms to Definition 1. The motivation is that neural networks can be universal approximators and the functional space parameterized by NN parameters leveraging L2O covers many popular point-based iterative optimizers such as Adadelta, Adagrad, Adam, AdamW, Adamax, NAdam, RAdam, RMSprop, Rprop, and SGD whose updates are functions of $z_t$.
>
> 3. *Why \hat{x} is computed using minimum rather than mean?*
>
> We provide the additional experiment comparing the choice of mean vs min in Table 8 and didn’t find a significant difference in uncertainty calibration.
>
> 4. *Why does UA-L2O fail in the seemingly simplest Rastrigin?*
>
> This is an important question that we need to clarify.  The definition of success or failure is not by optimization performance, as UA-L2O was designed for uncertainty calibration (being aware of the uncertainty in solutions) and not intended for optimization performance.  The failure means that the estimated confidence is off the actual chance that confidence intervals contain the global optima (*knowing* probabilistically whether optimization performance might be good). So the major assessment is in the middle column of Figure 3, that is, the accuracy of confidence estimation.  Since UA-L2O outperformed other methods in confidence estimation, even the right column (tightness of confidence intervals) is not comparable among methods.  As to the optimization performance, we are happy to see that they even improved against competitors in two of three functions, showing that uncertainty awareness could sometimes improve optimization performances.
>
> 5. *UQ dip in dim-27 of Ackley and Griewank*
>
> That is an interesting question that had also caught our attention earlier.  The dip could not be attributed to the higher dimensionality, as that was not the case for UA-L2O for other functions or/and other high dimensions.  A related observation was that DM\_LSTM also showed overconfidence in such exceptions, but the few exceptions cannot be attributed to L2O either.  We also note that the confidence intervals were very tight in such cases, which could be partially contributing to such overconfidence.
>
> 6. *what is $x_t^\phi$?*
> $x_t^\phi$ is the output of L2O with parameter $\phi$ at time step t (and therefore with trajectory $z_{t-1}$).
>
> 7. *Grammar mistakes, puzzling parts and typos*
>
> We thank the reviewer for pointing them out and sincerely apologize for the quality of writing.  We have made the corrections and some revisions.  And we pledge to further improve the presentation in the manuscript.
>
> We hope that the clarifications and the additional experiments above have addressed the reviewer’s concerns. And we look forward to hearing any feedback and addressing further comments.

---

> > ### Comment · Reviewer_Q3k1 · 2021-11-23
> > **Response to authors**
> >
> > Thank you very much for the detailed explanation.

---

> > > ### Author Response · Authors · 2021-11-29
> > > **Response to reviewer Q3k1**
> > >
> > > Thank you again for the helpful comments and suggestions!

---

### Official Review · Reviewer_x9M2 · 2021-10-28

**Correctness:** 2
**Technical Novelty And Significance:** 3
**Empirical Novelty And Significance:** 2
**Recommendation:** 5
**Confidence:** 3

**Main Review:**

In the very beginning, you say that no optimization algorithm can guarantee global optimum in general. It almost sounds like you say that your solution can. Do the authors agree with this?
Also, Bayesian optimization *in theory* can guarantee global optimization, but in practice probably not.

I would like the authors to give concrete ideas of how the space G looks. Does it contain usual optimizers such as Adam? And different levels of momentum etc.
Is it always a good idea to just make this space so high-dimensional?

On page 4 you say you extend the "Bayesian treatment". Can the authors expand on what they mean? Do you use more then Bayes theorem?

Why does MCMC lead to degenerate solutions? My belief is that MCMC in general is more accurate than variational inference, so is there something to worry about here?

The first equality in (4) is not the ELBO. The KL distance here is the difference between the ELBO and the marginal log likelihood.
Can the authors do the computations in the second equality for me?

Can I use the variation in g to conclude when convergence has happened?

You mention, but only very briefly, warm-starting theta. How is this done in practice? And how good is the solution immediately after warm-starting? In other words, how much improvement is done after this warm-start?

I found the experiment section very difficult to read, and therefore I am not convinced of the method's superiority over the compared methods.
Also, in future versions, I think a comparison with some hyper-parameter optimization is necessary.



**Summary Of The Paper:**

The paper introduces uncertainty-aware learning-2-optimize methodology. In particular, they parametrize a optimizer space as a neural network g and use variational inference to compute approximate posterior of g and the attained best solution x*.

**Summary Of The Review:**

There are some clear errors in the paper, and I am not convinced that the empirical performance justifies this methodology to be accepted at ICLR.

---

> ### Author Response · Authors · 2021-11-22
> **Response to reviewer x9M2**
>
> We thank the reviewer for the useful comments.  Although the reviewer’s summary mentioned “clear errors” and “empirical performance”, by going through the detailed comments we found that they are mostly for clarifications.  We understand that our original presentation presented barriers to understanding the motivation and interpreting the performance gains.  We make the clarifications below and in the revision and hope that they address the reviewer's concerns.  And we will continue improving the presentation of the manuscript.
>
> 1. *Clarifying motivations*
>
> By stating the fact that there is no general guarantee of global optimum for nonconvex optimization, we did not claim that we had a solution with a guarantee.  Rather, we used that fact to suggest the importance of uncertainty awareness for optimization solutions.  Meanwhile, as optimizers are directly responsible for optimization solutions and their uncertainty is not considered to date, we motivate our uncertainty-aware learning to optimize (UA-L2O), aiming at better uncertainty awareness by considering optimizer uncertainty.
>
> 2. *Detailing the optimizer space $\mathcal{G}$*
>
> The space of optimizers considered here is restricted to point-based iterative update rules, as described  in Sec. 3.1.  We leverage the parametrization of L2O (https://arxiv.org/abs/1606.04474) to construct the optimizer space. Concretely, we construct L2O with an LSTM taking the gradient trajectory as input as stated in Section 2. Since neural networks can be universal approximators and LSTM is a sequential encoder, the corresponding optimizer space parameterized using L2O covers many hand-crafted and popular point-based optimizers, such as Adadelta, Adagrad, Adam, AdamW, Adamax, NAdam, RAdam, RMSprop, Rprop, and SGD, whose update rules are functions of $z_t$.
>
> 3. *Clarifying Bayesian treatment*
>
> "Bayesian treatment” in the paper refers to treating the optimizer as a distribution rather than a prefabricated fixed one (Contributions on Page 1 and Technical Approach on Page 3), and accordingly, it did involve the Bayes theorem (Eq. 2 on Page 4).
>
> 4. *Why does MCMC lead to degenerate solutions?*
>
> We believe that this was not an intended claim and apologize if our presentation sounded like that anyhow.  What we meant was that, directly optimizing equation (3) which is maximum a posteriori (MAP) leads to degenerate solutions, and furthermore optimization via MCMC encounters computational intractability considering the high-dimensional optimizer space. This is now revised accordingly (“Training” on Page 4).
>
> 5. *Clarifying ELBO*
>
> We apologize for the confusion.  We didn’t mean that the first equality of Eq. (4) is ELBO.  What we meant to say was that, maximizing ELBO is equivalent to minimizing our objective, the left hand side of the first equality (https://arxiv.org/abs/1301.3838).  To minimize the misunderstanding, we have now revised the expression above Eq. (4) to be “we introduce the following objective function whose minimization is equivalent to maximizing the evidence lower bound (ELBO) (see Appendix D for the derivation)”.  For this equivalency claim and the second equality, we have included detailed derivations in Appendix D.
>
> 6. *Can I use the variation in $g$ to conclude when convergence has happened?*
>
> The reviewer has made an excellent suggestion.  We think that the posterior of the optimizer and ultimately the posterior of the optimization solution could potentially be used to judge empirical convergence (but no theoretical convergence is claimed here).  Further experiments are needed to examine the usefulness.

---

> > ### Author Response · Authors · 2021-11-22
> > **Response to reviewer x9M2 (2)**
> >
> > 7. *How was warm-starting done and how much improvement after warm-starting*
> >
> > The warm-starting of $\theta$ means that we first train our model in a non-Bayesian way as equation (1) and then use the resulting $\theta$ as the warm start for fine-tuning the mean variational parameters in the second stage (“Two-stage training” on Page 6).  As the reviewer requested, we provide the warm-start only performances in Table 10 of the Appendix.  Apparently, compared to just using the first stage, the two-stage UA-L2O much improved both the uncertainty calibration (our central goal) and the optimization quality (not our intended goal).
> >
> > 8. *Experiment section is difficult to read thus not convincing. Need comparison to HPO*
> >
> > We appreciate the suggestions.  We have revised the Intro to emphasize that “Being able to quantify solution uncertainty directly provides calibration with ensured awareness of the solution quality and usefulness (and another potential benefit is in optimization performance by enhancing the search efficiency)”.  We have revised the “Evaluation Metrics” to indicate that uncertainty calibration (the accuracy of confidence levels) is the major goal and interestingly optimization performance sometimes improves as well.
> >
> > In the experiment section, for optimizing test functions, we can see in the middle column of Figure 3 that UA-L2O consistently achieves better uncertainty calibration by matching estimated confidence levels and actual chances of “success” (corresponding confidence intervals contain the global optima).  Note that other methods, whenever having tighter confidence intervals, were often too overconfident to realize that the true global optimum were much less often inside such intervals than their confidence levels suggested.  For the application of privacy attack (Table 1), we can see that all competing methods but PSO and PSO-BO had less accurate confidence estimation; and PSO and PSO-BO had much looser confidence intervals, indicating a flat and uninformative posterior that led to worse optimization performances than UA-L2O.  For the application of  protein docking, UA-L2O improved against BAL, a state-of-the-art method for the application, in both confidence accuracy and optimization performance.  We have revised the Experiment section and we will continue improving the presentation.  We sincerely hope that the revision would lower the barrier from the presentation to appreciating the significance of our results.
> >
> > We hope that the clarifications and the additional experiments above have addressed the reviewer’s concerns. And we look forward to hearing any feedback and addressing further comments.

---

> > > ### Comment · Reviewer_x9M2 · 2021-11-27
> > > **Thank you**
> > >
> > > I thank the reviewers for their clarifications, and I think the new experiments and experiment section improves the paper greatly. I am changing my score to 5 after having read other reviews and discussions too.

---

> > > > ### Author Response · Authors · 2021-11-29
> > > > **Response to reviewer x9M2**
> > > >
> > > > We thank the reviewer for carefully going through our responses and revision as well as kindly reconsidering the score.  If there is any remaining concern, please do feel free to let us know.  Thank you again!

---

### Official Review · Reviewer_vJE3 · 2021-10-29

**Correctness:** 3
**Technical Novelty And Significance:** 3
**Empirical Novelty And Significance:** 3
**Recommendation:** 5
**Confidence:** 3

**Main Review:**

I found the main idea of the paper novel and interesting. The experimental section is extensive and shows the benefit of using the method across different applications. However I believe Section 3 of the paper is hard to follow and understand. Particularly, I think the authors should clarify the following points:

1) Provide an intuition for what the likelihood and the prior represent. What is $p(g|\mathbf{z}_t)$ that is  $p(g|\mathcal{D})$ representing? In standard Bayesian inference $p(\theta|\mathcal{D})$ gives how probable is $\mathcal{D}$ to be generated from $\theta$. Is it correct to say that in your setting $\mathcal{D}$ are the data points $(x_0….x_t)$ evaluated via optimization (and potentially the gradients) and $p(g|\mathcal{D})$ represent the probability of $g$ being the optimizer used to generate the data, ie used to optimize?
2) It is not clear to me why you are starting from an optimizer $g$ and the you speak about $\phi$. It seems to me that $g$ is a deterministic function parametrized by $\phi$ so I don't see the point of having both eq 2 and eq 3.
3) Definition 1. In point 2 there is no $\mathbf{x}$ on the left but the distribution on the right factorizes across $\mathbf{x}_i$. Are you making the assumption that $\mathbf{x}_i = \mathbf{z}_i$ or is $\mathbf{z}_t$ a set of historical information (iterates + gradients) as specified above?
4) How exactly are you sampling from (5)?
5) Could you specify the form of the assumed variational distribution $q(\cdot)$ in Section 3.2?
6) It seems like the paper is highly based on L2O, it would be useful to be more explicit about what you are improving with respect to that paper. Even thought that method has no uncertainty estimation it would be nice to have it in the experimental session? also, most baselines under comparison are non-Bayesian methods and the primary goal of the paper is to quantify uncertainty. Why are you not comparing to BO?
7) In the experimental session, in the paragraph "Evaluation metrics", $\mathbf{x}^*$ is sampled from $p(\mathbf{x}^*)$. Is this supposed to be $p(\mathbf{x}^* |\mathbf{z}_T)$ as given in Eq 5?
8) It is not clear to me why a lower value of $r_{\sigma}$ should be preferred? My understanding is that the bounds in Eq 6 are computed by sampling from Eq 5 and then getting the lowest value for $\hat{\mathbf{x}}$. The true value $\mathbf{x}_{true}$ is not involved thus we could have a narrower posterior but be off the true optimum.


**Minor comments**
1) What do you mean by “**Optimizing** the posterior via Markov chain Monte Carlo (MCMC)”? Do you mean sample from the posterior distribution? Or computing the MAP? I don't see how that would lead to a degenerate posterior distribution.
2) Figure 1 could be used to explain the methodology better. It is not mentioned in the text and the caption is to short to be informative.
3) It would be nice to include a discussion of why using a NN to model the optimizer is a good choice


**Summary Of The Paper:**

This paper considers the problem of quantifying the uncertainty of an optimizer used to do inference on a given model. The authors take a Bayesian approach and treat the optimizer as a random variable in the space of algorithms. They derive an approximate posterior via variational inference.


**Summary Of The Review:**

I really like the idea of considering the additional level of uncertainty and I find it novel. However, I found the methodological session difficult to follow and I think that could be improved by providing an intuition on the prior on $g$ and the connected posterior. At the practical level it is not clear to me how the proposed approach could improve the performance of existing algorithms.

---

> ### Author Response · Authors · 2021-11-22
> **Response to reviewer vJE3**
>
> We appreciate the reviewer’s interest in and suggestions to our manuscript.  Per the reviewer’s comments, we have made the following clarifications.
>
> 1. *Intuition for the optimizer prior, likelihood, and posterior*
>
> The reviewer’s understanding for the optimizer’s prior, likelihood, and posterior is correct.  We have now added the following explanations after introducing Definition 1: “Intuitively, the prior $p(g)$ represents the belief about well-performing optimizers, the likelihood $p(z_t | z_{t_0}, g)$ represents the probability of observing an optimization trajectory (data) under a given  optimizer $g$, and the posterior $p(g | z_t)$ represents the probability for $g$ being the optimizer generating the observed data.”
>
> 2. *Optimizer $g$ and its parameterization $\phi$*
>
> To target the uncertainty within the optimizer, $g$ is treated as a random sample in the space of iterative optimizers considered ($\mathcal{G}$), rather than a deterministic sample (such as a prefabricated Adam).  The optimizer space is parameterized by parameters of neural networks $\phi$, leveraging L2O, as the reviewer mentioned.  The reviewer is right that going from Eq. (2) to (3) is based on the parameterization and a Gaussian prior, and Eq. (3) is included for better understanding and easier notations for the objective function in Eq. (4).
>
> 3. *Notation in definition 1*
>
> The reviewer is right that $z_t$ represents a set of historical information (iterates + gradients).  As defined on Page 2, “We define $z_t$ as optimization trajectories'  historical information up to time $t$, e.g.,  the existing iterates $x_0,\ldots,x_t$, and/or their gradients $\nabla f(x_0),\ldots,\nabla f(x_t)$.” To remind readers of this notation, on Page 7 we have now included “$g \in \mathcal{G}$ produces an update $g(z_t)$  dependent on $z_t$, the current/past zero-th order and/or first-order information.”
>
> 4. *How exactly are you sampling from (5)?*
> 5. *The form of $q(\cdot)$ in Sec. 3.2*
>
> The form of the variational distribution $q(\cdot)$ is an L2O with LSTM backbone with Gaussian-parametrized weights (an analogy would be Bayesian Neural Networks using variational parameters mean and standard deviation to parametrize Gaussian weights). For posterior estimation, according to Eq. (5) we run the learned model for 10,000 times with random initialization and accordingly, different trajectories $z_t$.  The optimizer parameter $\theta$ is sampled from $q(\theta | \phi^*)$ once per iteration, following variational inference as in https://arxiv.org/abs/1312.6114 and https://openreview.net/forum?id=Sy2fzU9gl.  These contents are now included on Page 4 (at the end of Sec. 3).
>
> 6. *Missing baselines*
>
> “DM_LSTM” is actually the requested L2O without uncertainty estimation, which is now clarified on Page 5 (at the beginning of Sec. 4).
>
> At the request of the reviewer, we have included Adam and PSO with BO (see results in Fig. 3 and Table 1).  We have also newly included Adam with learning rate ensembles (Adam_lr_Ensemble) and Adam with stochastic gradient MCMC (Adam_Noisy_Gradient), as in Table 1.  UA-L2O outperformed these newly compared methods in confidence accuracy for nonconvex optimization.  It also outperformed them in optimization performance in two of three nonconvex optimization cases and in privacy attack.
>
> 7. *Should $p(x^\*)$ be $p(x^\* | z_T)$?*
>
> The reviewer is right.  $p(x^*)$ was meant to be a shorthand but that was not appropriate.  We have now revised it to $x^* \sim p(x^* | z_T)$  where $T$ is the maximum number of iterations (Page 6, “Evaluation Metrics”).
>
> 8. *Why the lower $r_\sigma$ is preferred?*
>
> The reviewer is right that tighter confidence intervals (lower $r_\sigma$) are only preferred when the confidence levels are accurate.   We have now clarified in “Evaluation Metrics”:
>
> “UA-L2O is designed for uncertainty awareness in the optimizer.  To assess the calibration of uncertainty quantification (UQ), we use the accuracy of confidence,  following Ortega et al. 2012 and Cao and Shen 2020.  In other words, we compare expected confidence levels $\sigma$ as in Eq. (6) (0.8 and 0.9 in this paper) and the empirical  probability $\varepsilon_\sigma$ of "success'' when the corresponding confidence intervals indeed contain the global optima.  When there is a tie in the major assessment metric $\sigma-\varepsilon_\sigma$,  the tightness of the confidence interval for the solution's proximity to the global optimum (Eq. 6), $r_\sigma = ub_{\sigma} - lb_{\sigma}$, serves as a tie-breaker.”

---

> > ### Author Response · Authors · 2021-11-22
> > **Response to reviewer vJE3 (2)**
> >
> > Minor 1. *Optimizing the posterior via MCMC*
> >
> > We apologize for the confusion.  The sentences are now revised to
> > “For maximum a posteriori (MAP) estimation of the optimizer parameters, directly maximizing the optimizer posterior in Eq. (3) via Markov chain Monte Carlo (MCMC) would encounter … “ By “degenerated point estimation” we mean that “a degenerate distribution and point estimation without uncertainty”.
> >
> > Minor 2. *Figure 1*
> >
> > We thank the reviewer for the suggestion.  We have now revised the text to better involve the figure for illustrating the concepts.
> >
> > We hope that the clarifications and the additional experiments above have addressed the reviewer’s concerns. And we look forward to hearing any feedback and addressing further comments.

---

> ### Author Response · Authors · 2021-11-29
> **Response to reviewer vJE3**
>
> Thank you again for the helpful comments and suggestions.  We are eager to hear from you any feedback to our response and we hope to receive and respond to the feedback during the discussion period which ends soon on Nov. 29.  Thank you!

---

### Official Review · Reviewer_3nHB · 2021-11-03

**Correctness:** 3
**Technical Novelty And Significance:** 3
**Empirical Novelty And Significance:** 2
**Recommendation:** 6
**Confidence:** 4

**Main Review:**

The paper is generally well-written and easy to follow. The idea of quantifying the uncertainty of the optimization algorithm is novel and interesting. However, there are some unclear points that make me not sure of the acceptance of the paper.

1) Motivation is not clear - why do we need optimizer uncertainties?

The "contributions" section in the introduction states that the paper discusses the fundamental question of 1) why modeling optimizer uncertainty, and in my opinion, this would be the most important question to be discussed. However, it is not clear from the paper why the optimizer uncertainty really matters.  At least, judging from the experiments, uncertainty-aware optimizers do not always give superior optimization results in terms of error, as can be seen from the results for the synthetic functions. UA-L2O seems to improve calibration, so the optimizer uncertainty somehow increases for the difficult optimization problems, and the authors stated in section 4.4 that "such calibration capability is critical to detect the failure cases in real-world application".  However, I don't think better calibration can actually lead to better detection of failure cases. For instance, in the synthetic experiments, the uncertainty calibration metrics displayed in the second and third columns do not show significant improvement over baselines without uncertainty; I mean UA-L2O is indeed better for some cases, but the margin is not that significant, so it is not clear we can actually make use of such uncertainty to actually detect the failure cases. In order to argue this, 1) one should first clearly define what a "failure" case is, and 2) do some experiments to run multiple optimization runs, and 3) do the binary classification based on the estimated uncertainty and compute AUROC to quantitatively show that we can actually benefit from the estimated uncertainty in terms of detecting failure cases. For the purpose of detecting failure cases, I can also think of very naive baselines, such as just monitoring the validation loss or training loss values and treating the ones with loss values larger than a certain threshold as a failure.

2) Important details are missing

As written in the review of the paper, L2O algorithms usually require a meta-training stage where the learned optimizer is trained with multiple optimization runs for the tasks sampled from some specific task distribution. Such meta-training is indeed time-consuming, and requires care in design; for instance, we need a diverse task distribution to guarantee the generalizability of the learned optimizers. However, I fail to see the detailed description of the meta-training protocols, other than the description of the ELBO being used for the inner optimization objective.  Also, it is not clear how the Monte-Carlo estimation is done for the actual deployment of the learned optimizer; how many samples are required? Are the samples drawn at the initial stage of the optimization or drawn at each iteration of the optimization? How sensitive is the algorithm to the number of samples? How does the algorithm scale with the number of samples being used? I particularly think the last question is important because it is directly related to the practicability of the algorithm; at least, I'd like to see the wall-clock time of the UA-L2O compared to non-Bayesian baselines.

3)  The "space of algorithms" is a slight over-statement

Although I find the notion of the "space of the optimization algorithms" quite appealing, the actual implementation given in the paper seems rather disappointing; as the authors already stated in the paper, there are many elements to be considered for a single optimization run. Needless to say the parameters $\phi$ for the learned optimizers, we also have the things like initialization scheme (or initial distribution for the parameters and their hyperparameters),  learning rate schedule, batch size, number of epochs to run, and so on. The uncertainty modeled for optimizer in this paper is restricted to the meta-parameter $\phi$, so I think to say that the proposed algorithm considers a "random sample of an optimizer $g$ from the space of optimizers $G$" is a slight over-statement.

4) Some important baselines are missing

The easiest baseline I can think of is ensembles; each run of an ensemble (with different random seeds) can be considered as a sample from an implicitly defined space of optimizers. Ensembles are known to excel in many applications both in terms of predictive accuracy and uncertainty calibration (although such uncertainty is for prediction, not like the optimizer uncertainty discussed in this paper). The downside of ensembles is that they require a longer training time than the usual optimization algorithms, but L2O also requires a considerably long meta-training time. So I think it is definitely worth comparing the proposed approach to (deep) ensembles with non-Bayesian optimizers. Another baseline I think worth comparing is stochastic gradient MCMC algorithms such as stochastic gradient Langevin dynamics or stochastic gradient Hamiltonian Monte-Carlo; such algorithms can be understood as noise-injected versions of gradient-based optimization algorithms, and samples collected from such samples can also be used to construct optimizer uncertainty (at least in terms of proximity to the true optima).

Considering the above-mentioned concerns, I recommend rejection, but I'm happy to discuss with the authors. Please let me know if I'm misunderstanding something.


**Summary Of The Paper:**

This paper proposes a Bayesian Learning to Optimize technique called Uncertainty-Aware Learning to Optimize (UA-L2O). The main contribution of this paper is its introduction of the concept of the space of optimization algorithms and treating an optimizer as a random sample from it, and deriving a proper Bayesian inference algorithm based on that concept. The space of optimization algorithms, to simplify, is expressed as a space of parameters of learned optimizers $\phi$, and a Gaussian prior is placed on those parameters. The posterior inference procedure is approximated with variational inference with reparameterization tricks, and the actual deployment of the learned optimizer is done with Monte-Carlo approximation with learned variational distribution for the optimizer parameters $\phi$. The proposed algorithm is applied to various synthetic and real-world optimization problems and showed some useful calibration behaviors.

**Summary Of The Review:**

The idea is interesting, but there are a few points to be addressed.

---

> ### Author Response · Authors · 2021-11-22
> **Response to reviewer 3nHB**
>
> We are thankful that the reviewer has found our ideas novel and interesting and has made inspiring comments and suggestions.  We do find that some parts of our earlier presentation have obstructed the understanding, including the motivation and Figure 3.  We thus make clarifications here and corresponding revisions in the paper.
>
> 1. *Motivation is not clear - why do we need optimizer uncertainties?*
>
> The goal of modeling optimizer uncertainty is for better uncertainty awareness of the optimization solutions, not for better optimization performance.  To minimize the confusion, we have revised the following sentence in the opening paragraph into "Being able to quantify solution uncertainty directly provides calibration with ensured awareness of the solution quality and  usefulness (and another potential benefit is in optimization performance by enhancing the search efficiency)."   And the rationale that considering optimizer uncertainty would help that goal is that the optimizer "is directly responsible for deriving the end solutions" yet "inconspicuous attention was paid to the uncertainty arising from the optimizer" (2nd paragraph of Introduction).
>
> For reasons above, the most important assessment metric is in calibration and not in optimization.  In other words, it is about whether the model-estimated confidence level ($\sigma$) matches the actual probability ($\epsilon_{\sigma}$) of a "success" when corresponding model-generated confidence intervals do contain global optima (otherwise a “failure” case is defined).  UA-L2O actually significantly outperformed all competing methods, including newly added baselines, in calibration (accuracy of confidence $|\epsilon_{\sigma}-\sigma|$) for nonconvex optimization, as seen in the middle column of Figure 3.   UA-L2O estimates confidence intervals from the posterior and its belief about the probability of such a success or failure (confidence level) was much more accurate compared to competing methods.
>
> Optimization.  We agree with the reviewer that better calibration (alone) does not guarantee better optimization. Instead we argue that better calibration and reliability is our goal and UA-L2O showed better calibration with similar or better optimization performances (better optimization in two of three test functions, privacy attack, and four of five protein docking cases).
>
> Confidence intervals.  As Reviewer vJE3 has correctly pointed out, the tightness of the confidence intervals $r_{\sigma}$ alone is not that meaningful for assessing calibration especially when the estimation of confidence levels $\sigma$ is off.  Given that competing methods had (much) worse accuracy in confidence levels compared to UA-L2O as seen in the middle column of Figure 3, their sometimes tighter $r_\sigma$ better not be overinterpreted.  In order not to blur the central message, we are considering only showing the right column of Figure 3 in the Appendix in order not to blur the message.
>
> The contents above are now reflected in the much revised “Evaluation Metrics” on Page 6.
>
> 2. *Important details are missing*
>
> We apologize for not providing enough details on these occasions and provide them below and in the revision.
>
> Meta training.  As the reviewer correctly pointed out, a key for L2O lies in not only diverse but also relevant tasks (optimizee).  We have now included the details for such data in Appendix B.  The three test functions are benchmark datasets for L2O (https://arxiv.org/abs/2103.12828) and in each case the meta-training data are from a broad family of similar functions as in https://arxiv.org/abs/1911.03787.  For Rastrigin, we used $f_D(x) = \lVert A x - b \rVert_2^2 - c \sum_{i=1}^D 10\cos(2\pi x_i ) + 10 D $ where $A,b,c$ are sampled from i.i.d normal distributions; for Ackley, we used $f_D(x) = -20\exp(-0.2\sqrt{0.5\lVert A x - b \rVert_2^2}) - c \sum_{i=1}^D \exp( \cos(2\pi x_i)/D) + \exp(1) +20$; and for Griewank, we used $f_D(x) = 1+ \sum_{i=1}^D\lVert A x - b \rVert_2^2/4000 - c \prod_{i=1}^D \cos(x_i)$.  For privacy attack and protein docking, such details are also provided in Appendix B.
>
> Deployment.  Besides what was included in the beginning of Sec. 4, we have now included these details right after Eq. (5): “Specifically, for estimating the solution posterior, the learned model is run 10,000 times with random initialization and different trajectories $z_t$ accordingly.  The optimizer parameter $\theta$ is sampled from $q(\theta | \phi^*)$ once per iteration, following variational inference as in VAE and beta-VAE”.  Additionally, we have tested UA-L2O’s sensitivity to sample sizes ranging from 100, 1000, to 10,000 (Table 7 in Appendix E) and its wall-clock comparison to non-Bayesian methods (Table 9 in Appendix E).  These results indicate that UA-L2O was relatively insensitive to the considered range of sample sizes and its efficiency was comparable to non-Bayesian methods.

---

> > ### Author Response · Authors · 2021-11-22
> > **Response to reviewer 3nHB (2)**
> >
> > 3. *The "space of algorithms" is a slight over-statement*
> >
> > We agree with the reviewer that the space of optimizers considered here is restricted to the iterative update rules, as acknowledged in Sec. 3.1.  Such optimizers are parameterized using neural networks, leveraging L2O.  With neural networks as universal approximators and LSTM as sequential encoder, the corresponding L2O space covers many hand-crafted and popular point-based optimizers, such as Adadelta, Adagrad, Adam, AdamW, Adamax, NAdam, RAdam, RMSprop, Rprop, and SGD, whose update rules are functions of $z_t$.  Learning rate scheduler is indeed included in such a space.  To minimize confusion, we have rephrased the mentioned expression twice, which now reads “random sample from an algorithmic space of iterative optimizers”.
> >
> > We appreciate that the reviewer has suggested an even wider and more representative space of optimizers, by additionally considering initialization scheme, batch size, and the number of epochs.  We are eager to further develop UA-L2O by considering a more representative space of optimizers.
> >
> > 4. *Some important baselines are missing*
> >
> > We appreciate the important suggestions.  We have now included learning-rate ensembles of Adam, a non-Bayesian optimizer, and stochastic gradient MCMC of Adam.  They were first introduced on Page 5 (beginning of Sec. 4) and their performances on privacy attack can be found on Page 7 (“Adam_lr_ensemble” and “Adam_Noisy_Gradient” in Table 1).  Two more baselines suggested by other reviewers, Adam and PSO with hyper-parameter Bayesian optimization (“Adam-BO” and “PSO-BO”), have also been included on Page 5 and 7 accordingly.  From Table 1 we can conclude that UA-L2O outperforms these newly added baselines in both confidence accuracy and optimization performance.
> >
> > We hope that the clarifications and the additional experiments above have addressed the reviewer’s concerns.  And we look forward to hearing any feedback and addressing further comments.

---

> > > ### Comment · Reviewer_3nHB · 2021-11-25
> > > **Response**
> > >
> > > I really appreciate the clarifications and I am sorry for the late response. I still have some unresolved questions.
> > >
> > > 1. Motivation: I get that the proposed framework provides an uncertainty of optimization run; my point is, how would that be useful? For instance, Bayesian neural nets provide well-calibrated predictions with properly quantified uncertainty estimates. The uncertainty itself tells us how reliable the predictions are but in order to be truly useful, there should be some downstream tasks that can actually benefit from such uncertainty estimates. Indeed, for classification, one can use the uncertainty for out-of-distribution detection, active learning, Bayesian optimization, and even for model-based reinforcement learning. I understand that this is probably the first research to develop the notion of optimizer uncertainty, so it may be harsh to ask for such diverse downstream tasks, but at least for the failure detection (which is considered to be the most straightforward downstream task), I don't expect the uncertainty quantified by the proposed approach be a good metric for it.
> > >
> > > 2. Scalability: thanks for providing the wall-clock time. However, it is quite surprising actually to see that the proposed method that draws a considerable number of samples at each iteration does not cause severe overhead. I guess the problems considered are relatively low-dimensional; would the proposed method still be scalable for the usual settings in deep learning, often involving several million parameters for optimization? Also, how would the meta-training scheme scale to such settings?

---

> > > > ### Author Response · Authors · 2021-11-29
> > > > **Response to reviewer 3nHB**
> > > >
> > > > We thank the reviewer for the prompt response and apologize for taking the time to run additional experiments before posting this response.  The reviewer has made excellent follow-up comments.  Please see our responses below:
> > > >
> > > > 1. *Motivation*
> > > >
> > > > We appreciate that the reviewer has further clarified and suggested how the “usefulness” of uncertainty quantification should be tested.  At the suggestion of the reviewer, we have performed the experiment of “failure detection” as a basic downstream task.
> > > >
> > > > Specifically, an optimization solution is defined as “failure” if its distance to the global optimum is beyond a threshold $0.5\sqrt{d}$ (where $d$ is the dimension and the threshold was chosen for approximately balanced success/failure cases).  Instead of fixing the confidence level $\sigma$ and estimating the confidence interval $r$ (as observed in Figure 4 that better optimization correlated better with tighter confidence intervals), here we let each method estimate the confidence level $\sigma$ at fixed $r<0.5\sqrt{d}$ (“success”) for the optimization solution in each trajectory and use $\sigma$ to classify the quality of such solutions over 1000 trajectories. The classification performance was assessed using the area under the receiver-operator characteristic curve (AUROC) and the area under the precision-recall curve (AUPRC).
> > > >
> > > > The results for privacy attach are summarized as follows:
> > > >
> > > > || Adam | Adam_BO | PSO | PSO_BO | BAL | DM_LSTM | UA-L2O |
> > > > |--------------|--------------|-----------|------------|------------|------------|------------|------------|
> > > > | AUROC        | 0.53 | 0.59 | 0.74 | 0.78 | 0.00 | 0.37 | 0.99
> > > > | AUPRC       | 0.97 | 0.98 | 0.73 | 0.90 | 0.00 | 0.95 | 0.99
> > > >
> > > > 2. *Scalability*
> > > >
> > > > The reviewer was right that the previous response was based on the privacy attack experiment where the dimensionality was as low as 9.  We have now included a more extensive study of the wall-time versus dimensionality ranging from 10 to 2,000, using the optimization of Rastrigin function.  We report UA-L2O’s wall-time for both meta-training and meta-testing using a single GPU NVIDIA A100, the latter of which does not require gradient backpropagation.
> > > >
> > > > |Dimension| Meta-training (s)|Meta-test (s)|
> > > > |--------------|--------------|-----------|
> > > > |10|607|36|
> > > > |100|610|38|
> > > > |200|662|37|
> > > > |500|1,059|38|
> > > > |1,000|1,998|38|
> > > > |2,000|3,904|38|
> > > >
> > > > The results above show that the meta-training time did grow almost linearly with the dimensionality when dimensionality $>$ 200 (which is partly attributed to the fact that the number of samples is now adopted as a value correlated with the dimension).  So apparently meta-training in millions of dimensions for deep-learning model parameters is expensive yet still manageable, especially using multiple GPU cores.
> > > >
> > > > However, most importantly, meta-testing was much faster and the time remained flat with regards to the dimensions.  In other words, once UA-L2O is trained, its deployment remains fast and scales well to the dimensionality.  In fact,  L2O usually takes the setting of offline training (which could be time-consuming) and online deployment (which is fast enough and scales well) (https://arxiv.org/abs/2103.12828v1).

---

> > > > > ### Comment · Reviewer_3nHB · 2021-11-29
> > > > > **Thanks for the response**
> > > > >
> > > > > I really appreciate the additional results. They partially resolve my concerns and help me understand the usefulness of the algorithm. I raise my score to 6.

---

### Author Response · Authors · 2021-11-22
**General response**

We appreciate all the thorough and thoughtful comments from the reviewers.  When responding to the comments we realize that our presentation has presented a barrier to understanding the motivation of the study and interpreting the significance of the results.  We thus sincerely thank all the reviewers for spending the efforts seeing the value of our work through the writing.

In the responses to individual comments, we have made clarifications on the motivation, the major assessment metrics, and the significance of the results.  We have also included the following additional experiments:

- Additional baselines: HPO via Bayesian optimization baseline for both Adam and PSO;
- Additional baselines: ensemble and stochastic gradient MCMC for a non-Bayesian method (Adam);
- Ablation on UA-L2O with warm-starting only;
- Sensitivity of UA-L2O to different sample size;
- Ablation on $\hat{x}$ computed by mean versus minimum;
- Comparison of optimization wall-clock time.

The responses and new results are also reflected in the revision of the abstract, the opening paragraph, part of the Methods section, the Experiments section, and the appendices.  We thank the reviewers again for helping us improve the manuscript!

---

### Decision · Program_Chairs · 2022-01-20

**Decision:**

Accept (Poster)

**Comment:**

While the reviewers place this manuscript right at the threshold of acceptance, I find the revisions that they have made to address the majority of the reviewers concerns. That, combined with some of the reviewers' scores being slightly miscalibrated with their (largely positive) reaction to the author feedback, I am advocating for this paper to be accepted.